# Dynamics of the blood plasma proteome during hyperacute HIV-1 infection

Jamirah Nazziwa [1,2], Eva Freyhult[3], Mun-Gwan Hong [4], Emil Johansson [1,2], Filip Årman[5], Jonathan Hare[6,7,8], Kamini Gounder[9,10,11], Melinda Rezeli[5,12], Tirthankar Mohanty [13], Sven Kjellström [5], Anatoli Kamali[7,8], Etienne Karita[14], William Kilembe[15], Matt A. Price[7,8,16], Pontiano Kaleebu[17], Susan Allen[14,15,18], Eric Hunter [14,15,18], Thumbi Ndung'u[9,10,11,19], Jill Gilmour[20], Sarah L. Rowland-Jones [21], Eduard Sanders [22,23,24], Amin S. Hassan [1,2,23,25,26] & Joakim Esbjörnsson [1,2,21,26] ✉

The complex dynamics of protein expression in plasma during hyperacute HIV-1 infection and its relation to acute retroviral syndrome, viral control, and disease progression are largely unknown. Here, we quantify 1293 blood plasma proteins from 157 longitudinally linked plasma samples collected before, during, and after hyperacute HIV-1 infection of 54 participants from four sub-Saharan African countries. Six distinct longitudinal expression profiles are identified, of which four demonstrate a consistent decrease in protein levels following HIV-1 infection. Proteins involved in inflammatory responses, immune regulation, and cell motility are significantly altered during the transition from pre-infection to one month post-infection. Specifically, decreased ZYX and SCGB1A1 levels, and increased LILRA3 levels are associated with increased risk of acute retroviral syndrome; increased NAPA and RAN levels, and decreased ITIH4 levels with viral control; and increased HPN, PRKCB, and ITGB3 levels with increased risk of disease progression. Overall, this study provides insight into early host responses in hyperacute HIV-1 infection, and present potential biomarkers and mechanisms linked to HIV-1 disease progression and viral load.

While the blood plasma proteome typically remains stable in healthy individuals, perturbations have been documented in response to different infections, including severe acute respiratory syndrome coronavirus 2, and bacterial and viral pneumonia[1,2]. Understanding the differential expression of plasma proteomics in these infections played a pivotal role in informing diagnostic, prophylactic, and therapeutic interventions[3,4]. In HIV-1 infection, virus-host interactions during the earliest stages of infection—the hyperacute HIV-1 infection (hAHI, defined as the period from onset of plasma viremia to peak viral load)—trigger a complex network of cellular and tissue signaling events. This results in rapid systemic immune activation and reorganization of cellular microenvironments[5–8]. A notable aspect of this immune response is the cytokine storm, which is often accompanied by acute retroviral syndrome (ARS) in some individuals[9–12]. These inflammatory events contribute to substantial CD4 + T-cells loss and germinal center disruption, playing a critical role in shaping HIV-1 disease pathogenesis[6,8,13–15]. Indeed, disease progression to AIDS in HIV-1 infected individuals varies greatly, from a few months to several decades, and events during hAHI have been suggested to significantly influence the rate of disease progression[16–18]. In blood, HIV-1 viraemia becomes detectable about a week after infection, reaching a peak viral load (VL) of millions of virus particles per milliliter plasma 3–4 weeks after infection[5,6]. After the peak VL, the viraemia gradually decreases and stabilizes at a set-point level ~30–65 days after infection[5,6].

While pro-inflammatory and antiviral cytokines and chemokines have been studied extensively, longitudinal studies investigating blood plasma proteins during hAHI are lacking[13,19,20]. Recent advances in data-independent acquisition mass spectrometry (DIA-MS) based proteomics have made it possible to simultaneously identify and quantify thousands of proteins in plasma across a large dynamic range, significantly enhancing the discovery of potential biomarkers and regulators of disease[21]. Furthermore, the human protein atlas has mapped the cellular and tissue expression patterns of these proteins, facilitating analysis of proteome dynamics in response to infections[22]. In this study, we quantify the dynamics of the blood plasma proteome before, during and after hAHI, and present multiple plasma proteins associated with ARS, viral load responses, and HIV-1 disease progression.

## Results

### Study participants

Overall, 54 participants from Central and East Africa (International AIDS Vaccine Initiative [IAVI] cohort, $n = 39$), and South Africa (Durban cohort, $n = 15$) were included in the study (Fig. 1)[23–26]. The participants contributed 157 longitudinally linked plasma samples from three time points including Visit 0 (V0), collected at a median of 62 days prior to HIV-1 infection (interquartile range [IQR] 28–106 days); V1, median 10 days after HIV-1 infection (IQR 10–14); and V2, median 31 days after HIV-1 infection (IQR 28–37)[27]. Most participants were male ($n = 34$, 63%), aged below 25 years ($n = 32$, 59%), from Kenya ($n = 32$, 59%), infected with HIV-1 sub-subtype A1 ($n = 31$, 57%), and identified as men who have sex with men (MSM, $n = 28$, 52%, Table S1). Both age and sex are well-known factors influencing HIV-1 progression, whereas cohort and HIV-1 subtype may reflect potential variability due to demographic and virological differences[28]. Age, sex, cohort, and HIV-1 subtype were therefore assessed as potential confounders. The assessment showed that the IAVI cohort predominantly consisted of HIV-1 sub-subtype A1 infected male participants, whereas the Durban cohort exclusively was composed of HIV-1 subtype C infected female participants (Table S1). To avoid over-parameterization, all subsequent models were only adjusted for age and cohort (except for protein expression associated

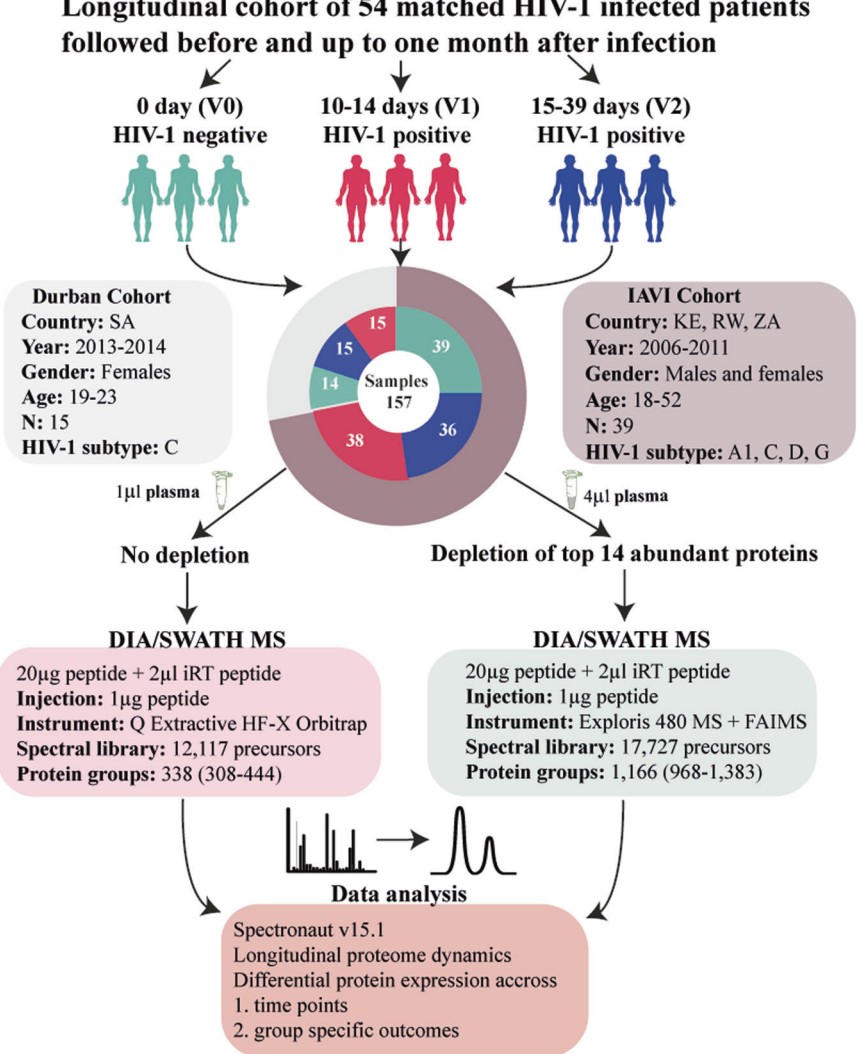

**Fig. 1 | Characteristics of the study participants.** The flowchart outlines the longitudinal sampling and proteomic workflow employed in the study. Fifty-four individuals from two distinct geographical regions provided three matched samples each. Plasma samples were prepared both with and without depletion of the top 14 most abundant proteins. These samples were then analyzed using Data-Independent Acquisition (DIA)/SWATH LC-MS/MS. Arrows indicate the flow of samples through each stage of processing, including plasma preparation, mass spectrometry analysis, and subsequent computational analyses, which were used to explore protein dynamics over time. Abbreviations: V0 visit 0 (collected before estimated date of infection), V1 visit 1 (collected 10–14 days post estimated date of infection), V2 visit 2 (collected 15–42 days before estimated date of infection), IAVI International AIDS Vaccine Initiative, SA South Africa, KE Kenya, DIA data-independent acquisition, SWATH sequential window acquisition of all theoretical mass spectra, MS mass spectrometry.

with ARS that was only adjusted for age, since ARS data was only available for the IAVI cohort). Eighteen of the 54 participants possessed protective HIV-1 HLA I alleles including B*58:01, B*57:02, B*57:03, and A*74:01, while 11 had disease-susceptible HLA I alleles including B*58:02 and B*18:01 (Table S2). However, no associations between HLA and HIV-1 control ($p > 0.05$) or disease progression were found ($p > 0.05$).

## Plasma proteome dynamics during hyperacute HIV-1 infection

To maximize protein detection, each sample was analyzed both in its neat form and following the depletion of the 14 most abundant proteins (Fig. 1, Supplementary Information)[29]. In total, 1549 protein profiles were detected, with 213 excluded due to absence in more than 80% of the samples (Fig. S1). Of the remaining 1336 proteins, 379 were detected in neat plasma samples, and 957 in depleted plasma. Among 1028 identified proteins with unique UniProt IDs and canonical protein form, 242 proteins had been previously classified as secreted proteins actively released into blood plasma, while 618 proteins were categorized as intracellular or tissue leakage proteins originating from tissues or dying cells. The remaining proteins were either secreted to other locations or represented immunoglobulin genes[22,30]. To determine the longitudinal within-host dynamics across V0, V1, and V2 for each of the 1336 identified plasma proteins, kmeans and hierarchical clustering were used. The analysis suggested six different longitudinal expression profiles (Fig. 2a). Of these, two demonstrated a significant decline during hAHI with a rebound to pre-infection levels after hAHI (referred to as "Rapid decrease-rapid increase", and "Slight decrease-rapid increase"); two demonstrated a decreased levels during and after hAHI ("Gradual decrease", and "Rapid decrease-slight increase"); one demonstrated an increase during hAHI followed with a decline to pre-infection levels after hAHI ("Rapid increase-rapid decrease"); and one demonstrated a sustained increase during and after hAHI ("Persistent increase"). Population-based protein dynamics, based on mean protein intensities, were analyzed and revealed distinct functional groupings. Proteins in "Rapid decrease-rapid increase" group, accounting for 7% of the determined proteins, were primarily involved in cell adhesion and extracellular structure organization. In the "Gradual decrease" group, which comprised 32%, proteins were typically involved in activation of immune response, cytoskeleton organization, protein transport and regulation of apoptosis. The "persistent" category, representing 18%, included proteins associated with epithelial development, complement activation, cellular extravasation, and viral entry into host cell (Fig. S2).

Next, significantly differentially expressed proteins between the study visits at the individual level were identified. An explorative principal component analysis showed distinct protein expression patterns by cohort and visit, with Durban participants clustering separately from IAVI participants (Fig. S3). To account for this potential confounder, we applied a linear mixed-effects model that included age, cohort, visit, and the interaction between cohort and visit, while also adjusting for principal components (PC1 and PC2) and patient-specific random effects. Using this model, we identified 97, 168 and 149 differentially expressed proteins between visits V1–V0, V2–V0, and V2–V1, respectively (Fig. 2b). Of these, two (V1–V0), 18 (V2–V0), and eight (V2–V1) proteins exhibited substantial changes with a log2 fold change (Log2FC) > ±1 (Fig. 2c; Table S3).

Among the 97 differentially expressed proteins at V1–V0, 51 proteins were upregulated, including 28 classified as secreted to blood and 23 as leakage proteins based on human protein atlas. Upregulated proteins at V1 were overrepresented for gene ontology biological process (GO-BP) terms, related to innate immune responses, stress responses, adaptive immune responses, transport, protein maturation, and regulation of vesicle-mediated transport (Fig. 2d). Of the 46 downregulated proteins at V0, 18 (40%) were classified as proteins secreted to blood. Over-representation analysis showed that 13 proteins were linked to GO-BP terms such as regulation of hydrolase/

peptidase activity and establishment of endothelial barrier (Fig. 2e). Furthermore, tissue-specific transcriptional signatures in V1–V0 indicated significant activation of tissue damage markers associated with esophagus mucosa and heart (Fig. S4)[31].

When comparing V2 with V0, 77 proteins were upregulated, whereof 58 (75%) were classified as leakage proteins and 39 upregulated were linked with various GO-BP terms, including blood coagulation, fibrin clot formation, protein maturation, positive regulation of cell-substrate adhesion, metal ion responses, positive regulation of substrate adhesion-dependent cell spreading, and cell adhesion (Fig. 2d). Of the 95 downregulated proteins at V2, 48 (51%) were classified as leakage proteins, and 53 were associated with GO-BP terms related to regulation of peptidase activity, inflammatory responses, stress responses, wound healing, immune responses, cytokine production of the Tumor necrosis factor superfamily, and neutrophil chemotaxis. Moreover, analysis of tissue-specific transcriptional signatures indicated that damage markers were activated in muscle skeletal tissues and lungs, and at the same time suppressed in whole blood, suggesting substantial tissue damage and cellular stress responses between V0 and V2 (Fig. S4).

Finally, 149 proteins were differentially expressed between V1 and V2, whereof 86 were upregulated. Based on over-representation analysis, 28 of these proteins (33%) related to gene expression, protein maturation, and macromolecule biosynthetic process. Of the 86 downregulated proteins, 53 were associated with inflammatory, stress, and innate immune responses, as well as detoxification, responses to biotic stimulus, complement activation, and neutrophil chemotaxis (Fig. 2e). Further analysis showed that proteins upregulated between V2-V1 associated with damage signatures in adipose, cervical, muscle skeletal, and lung tissue; whereas downregulated proteins were associated with tissue damage in skin and whole blood (Fig. S4). Overall, these findings indicate that HIV-1 infection induces a myriad of dynamic changes in protein expression that go beyond immune responses, influencing various biological processes such as cell mobility, metabolic function, and apoptosis.

## ZYX, SCGB1A1, and LILRA3 levels are associated with ARS

Data on AHI symptoms in the Durban cohort were missing. Hence, associations between protein dynamics and ARS were only conducted for participants from the IAVI cohort ($n = 33$, Fig. 3a). ARS was determined by latent class analysis based on the 11 symptoms including fever, headache, myalgia, fatigue, anorexia, pharyngitis, diarrhea, night sweats, skin rash, lymphadenopathy, and oral ulcers, as previously described[9]. The analysis suggested that 20 of the 33 participants (61%) had ARS. Participants with ARS had significantly higher prevalence in nine of the eleven symptoms than those without ARS ($p < 0.05$; Fisher exact test, Fig. 3b). Partial Least-Squares Discriminant Analysis (PLS-DA) was used to identify proteins collectively associated with ARS. The PLS-DA model that considered differences between V1–V0 and V2–V0 demonstrated the highest average performance measures in predicting ARS across 50 test sets (Table S4). This model had an average accuracy of 78% (as assessed through cross-validation), an Area Under the Receiver Operating Characteristic Curve of 82%, and a misclassification error of 20% (Fig. 3c). The PLS-DA analysis suggested 20 differentially expressed proteins, with variance importance scores >2, as potential indicators of ARS (Fig. 3d, e). Approximately half of these proteins at V1–V0 have been shown to be actively secreted into plasma, and primarily involved in regulation of inflammatory responses, immunity, and host-virus interactions. Proteins identified at V2–V0 were predominantly classified as tissue leakage proteins associated with cell motility and signaling (Table S5).

Next, we used linear mixed-effects models to assess how ARS was associated with protein expression over time. The analysis showed that three of the seven V1–V0 proteins that were determined as strong potential indicators of ARS in the PLS-DA analysis were also

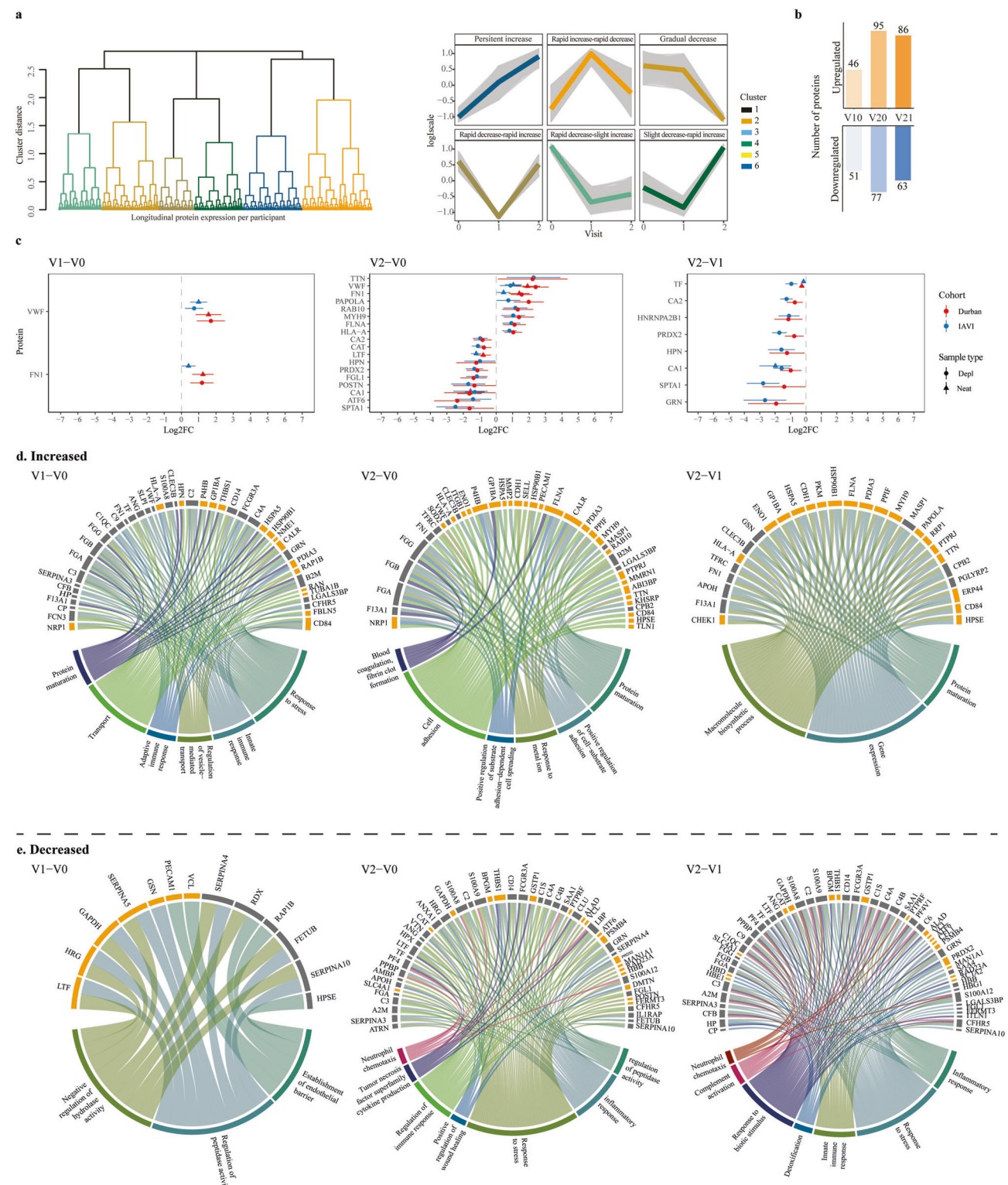

individually associated with ARS (Fig. 3e, f). More specifically, the average protein intensities of Zyxin (ZYX, a cytoskeleton protein, modulates inflammatory response in endothelial cells) and Secretoglobin family 1A member 1 (SCGB1A1, a pulmonary surfactant anti-inflammatory protein, plays an anti-inflammatory role in the lungs) were both ~8 times lower ($p < 0.005$) in participants with ARS compared to those without ARS (Supplementary Data 1)[32,33]. In contrast, the Leukocyte immunoglobulin-like receptor subfamily A member 3 (LILRA3, a soluble protein secreted by monocytes and macrophages and serves as an immunoregulatory receptor, balancing immune response/inflammation during infection) was approximately two times higher in participants with ARS compared to those without ARS[34]. Moreover, previous studies have indicated that ZYX can influence HIV-1 replication[35,36]. Collectively, these findings highlight a strong connection between inflammation, innate immunity, and cell motility with the manifestation of ARS.

## NAPA, RAN, and ITIH4 levels are associated with HIV-1 control

The period during which the VL was monitored varied between study participants. The median follow-up time was 4 years after the

**Fig. 2 | Acute HIV-1 infection alters the human plasma proteome.** Longitudinal protein expression profiles were investigated during hAHI. A comprehensive analysis of 83,643 protein combination values from all three-time points was conducted across 54 study participants, resulting in a total of 1336 profiles. **a** To identify the optimal clusters representing the longitudinal expression profiles for different groups, the elbow method was employed, leading to identification of six distinct clusters. These clusters were color-coded and plotted, with the x-axis denoting the visit number and the y-axis representing the scaled log-intensity per patient. **b** Bar plot illustrating the comparison in the number of differentially expressed proteins across visit differences. The height of each bar corresponds to the number of proteins while the bar color varies depending on the visit difference. **c** Forest plots indicating effect sizes (log2 fold change) and 95% confidence intervals for proteins significantly differentially expressed at 2 weeks and 1 month post estimated date of infection (EDI), relative to pre-infection levels (V1-V0 and V2-V0, respectively), as well as the difference between 2 weeks and 1 month post EDI (V2-V1). Effect sizes are shown in red for the Durban cohort, and blue for the IAVI cohort. Circles and triangles indicate depleted (depl) and neat plasma, respectively.

The statistical analysis was conducted using linear mixed-effects models with a random intercept for each patient, treating visit number as a categorical variable. The differential protein expression was assessed using a global ANOVA, with post hoc tests identifying specific visit comparisons (e.g., V0 vs. V1, V0 vs. V2, and V1 vs V2). The Benjamini-Hochberg's FDR method with a 5% FDR threshold was used to correct for multiple testing, with a fixed $p$-value cut-off of 0.005. **d, e** Circos plots visualizing the differentially expressed proteins from different visit differences in a circular layout. The lower ring represents the GO-biological processes associated with the proteins belong to, with each process color-coded for easy identification. The upper rings depict specific classification of these proteins, with proteins secreted in blood shown in orange and the tissue leakage proteins shown in gray. Abbreviations: V0 visit 0 (collected before estimated date of infection); V1 visit 1 (collected 10-14 days post estimated date of infection); V2 visit 2 (collected 15-42 days before estimated date of infection); V1-V0 difference between visit V1 and V0; V2-V0 difference between visit V2 and V0; V2-V1 difference between visit V2 and V1; Log2FC log 2-fold change. Source data are provided as a Source Data file.

estimated date of infection (EDI). To identify plasma proteins associated with HIV-1 control, 9 study participants that either initiated antiretroviral treatment (ART) within the first year of infection, or were followed for less than 1 year were excluded from the analysis (Fig. 4a). The remaining 45 study participants typically showed high levels of viremia following HIV-1 infection, which then subsequently decreased and reached a viral load set-point after ~50 days post-EDI (Fig. 4b). The median peak VL was 6.0 (IQR 5.2–6.4) log10 copies/ml at a median of 19 (IQR 15-31) days post EDI, and the median nadir VL was 4.5 log10 copies/ml at a median of 64 (IQR 43–73) days post EDI. Hierarchical clustering was used to group the 45 study participants into two clusters based on their VL profiles during the first year of infection (Fig. 4c). The clusters were defined as viral controllers (VL generally below 10,000 copies/ml for 12 months without ART, $n = 15$), and non-controllers (VL generally above 10,000 copies/ml for 12 months without ART, $n = 30$, Fig. 4d). No clinical parameters, or ARS were associated with viral control (Fig. 4e, Table S6).

Next, linear mixed-effects models were used to determine how viral load influence protein expression over time. The Alpha-soluble N-ethylmaleimide-sensitive factor attachment protein (NAPA, a protein involved in membrane fusion and vesicle trafficking within cells, and viral release in HIV by interacting with the HIV Gag protein to promote budding and efficient viral particle assembly) was more than twice as high among VL controllers compared with non-controllers at both V1-V0 and V2-V0 ($p < 0.005$, Fig. 4f, Table S7)[37]. Moreover, the GTP-binding nuclear protein Ran (RAN, a small GTP-binding protein primarily involved in the regulation of nucleocytoplasmic transport, mitotic spindle assembly, and nuclear envelope formation, that has been shown to be essential for the nuclear import of the HIV-1 pre-integration complex and integration of viral DNA into the host genome) was approximately three times as high among VL controllers compared with non-controllers at V2-V0 and V2-V1 (Supplementary Data 2)[38–40]. Finally, the Inter-alpha-trypsin inhibitor heavy chain H4 (ITIH4, an acute-phase protein involved in the stabilization of the extracellular matrix and regulation of inflammation that has be suggested to contribute to immune activation, tissue damage and disease progression in HIV-1 infected individuals) was approximately four times lower among VL controllers compared with non-controllers at V2–V0[41,42]. Collectively, these results indicate a strong relationship between viral control and virus-host interactions, particularly related to membrane fusion and nuclear import.

### HPN, PRKCB, and ITGB3 levels are associated with HIV-1 disease progression

Disease progression was defined using CD4 + T-cell responses. Participants contributed a median of 13 (IQR: 10-18) CD4 + T-cell count observations during the 12 months of follow-up after EDI (median

CD4 + T-cell count 520, IQR: 404–657 cells/mm³). Fast progressors were defined as participants that reached a CD4 + T-cell count <500 cells/mm³ within 12 months from EDI (excluding measurements within the first 6 weeks), whereas slow progressors were defined as participants who maintained CD4 + T-cell counts >500 during the same period. Among the 54 participants, 12 (22%) were classified as slow progressors and 42 (78%) as fast progressors (Fig. S5a). Study participants from Durban had a higher likelihood of slower disease progression compared with IAVI study participants ($p = 0.08$, Log-rank test, Fig. S5b). A Cox regression model, controlling for age and cohort, was used to compare changes in protein levels between visits. Seven, six, and 14 proteins were associated with faster HIV-1 disease progression at V1-V0, V2-V0, and V2-V1, respectively ($p < 0.005$, Fig. 5c, Table S8). Specifically, increased levels of Hepsin (HPN, HR = 1.4, CI = 1.1–1.7), Protein Kinase C Beta (PRKCB) (hazard ratio (HR) 1.3, CI = 1.1–1.6), Corticotropin Releasing Hormone Binding Protein (CRHBP, HR = 1.2, CI = 1.1–1.3), Proteasome subunit beta type-6 (PSMB6, HR = 1.3, CI = 1.1–1.5), Thioredoxin domain-containing protein 5 (TXNDC5, HR = 1.2, CI = 1.1–1.4), and Apolipoprotein C4 (APOC4, HR = 1.3, CI = 1.0–1.2) at V1–V0 were associated with an increased risk of HIV-1 disease progression. Decreased levels of Glutathione S-Transferase Mu 2 (GSTM2) at V1-V0 was associated with a faster disease progression (HR = 0.9, CI = 0.8-1.0). For clarity, an HR of 1.3 indicates that a doubling of the protein level (a 1-unit increase on log2 scale) from V0 to V1 was associated with a 30% increased risk that CD4 + T-cell counts drops below 500 cells/µl blood within 1 year from EDI. Interestingly, according to the HIV-1 interaction database, all the identified proteins have been shown to interact with the HIV-1 envelope (Supplementary Data 3)[35]. Moreover, increased levels of Integrin subunit beta 3 (ITGB3, HR = 1.3, CI = 1.1–1.5), Heat shock 70 kDa protein 8 (HSPA8, HR = 1.1, CI = 1.0–1.2), D-dopachrome tautomerase like protein (DDTL, HR = 1.2, CI = 1.1–1.4) and Ubiquitin B (UBB, HR = 1.1, CI = 1.0–1.2) at V2-V0 were associated with an increased risk of disease progression. Decreased levels of CD84 (HR = 0.9, CI = 0.8–0.9) and Latent-transforming growth factor beta-binding protein 1 (LTBP1, HR = 0.7, CI = 0.6–0.9) were associated with decreased risk of disease progression. Notably, all these proteins interact with different HIV-1 proteins to mediate (Supplementary Data 3).

### Longitudinal protein dynamics in hyperacute HIV-1 infection
Finally, a sliding window approach was used to generate spline curves reflecting the population-based average dynamics of identified key proteins in blood plasma during hAHI (Fig. 6, Supplementary Data 4). The protein levels were plotted relative to pre-infection levels for each study participant on the day of sample collection post EDI. Key proteins associated with ARS, viral control, and disease progression were selected based on the above analyses. For clarity, proteins with similar

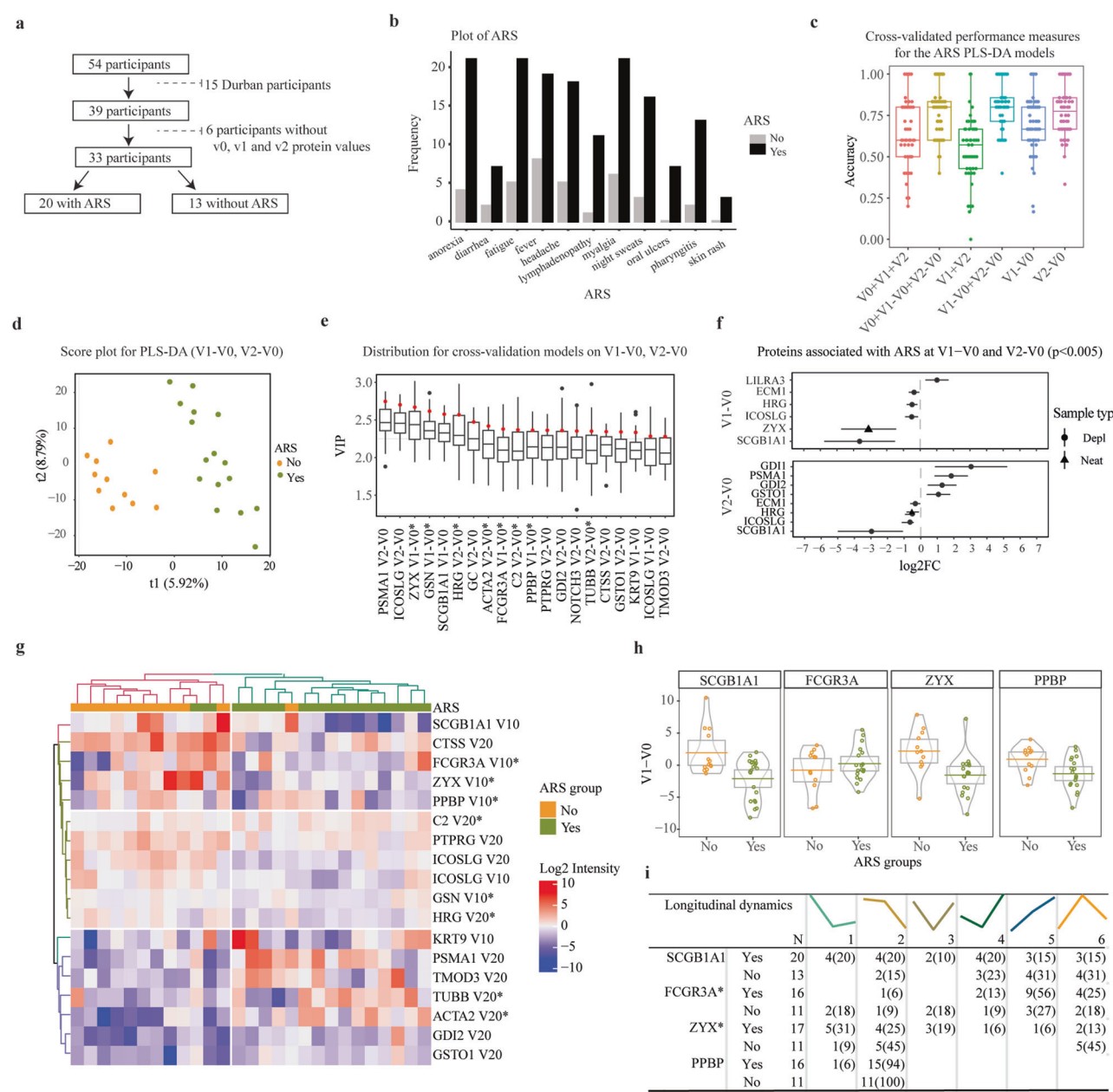

dynamics in the group comparisons were excluded from this analysis. The analysis showed an exceptional variability in the trajectory of the different proteins over time and indicated that both the dynamics of overexpressed proteins (defined as *stormers*) and underexpressed proteins (defined as *slumpers*) were common during hAHI.

## Discussion

Our study represents the largest and most comprehensive longitudinal MS-based study of protein dynamics in hAHI. Moreover, determination of pre-infection protein levels for each study participant enabled analysis of relative changes in protein expression close to the HIV-1 transmission event. The analyses showed significant alterations of the plasma proteome during hAHI, and most changes were transient as the infection progressed. Specifically, elevated expression of SCGB1A1 and ZYX during hAHI and decreased levels of LILRA3 were associated with the absence of ARS; increased levels of NAPA and RAN during hAHI, and decreased levels of ITIH4 after hAHI were associated with lower viremia; and increased levels of HPN and PRKCB during hAHI, and ITGB3 and DDTL after hAHI, were associated with faster disease

progression. These findings are particularly interesting since they represent host factors altered during hAHI that to the best of our knowledge have not been linked to ARS and HIV-1 pathogenesis in the past. Previous longitudinal studies in hAHI have focused on the dynamics of pro-inflammatory and antiviral cytokines and chemokines, particularly those immune markers that exhibit increased levels and thereby part of the well-described cytokine storm[9,19,20]. Strikingly, our large-scale analysis of the blood plasma dynamics in hAHI indicated that the differentially expressed proteins associated with ARS, VL, and disease progression was overexpressed (here defined as *stormers*, in analogy with the cytokine storm) to a similar extent as being underexpressed (here defined as *slumpers*). This warrants further studies, and it is possible that potential biomarkers and treatment targets will be identified among slumpers to a similar extent as among stormers (which have been the focus so far). Moreover, our study expands beyond studies of acute phase and inflammatory proteins and includes proteins involved in antigen presentation, cell transport, proteolysis, and cytoskeleton modulation during hAHI. For example, vWF and FN1 were the most significantly elevated proteins during hAHI

**Fig. 3 | Zyxin, Secretoglobin family 1A member 1, and Leukocyte immunoglobulin-like receptor subfamily A member 3 are associated with ARS.** **a** Flow chart representing the total number of samples used in ARS classification and the exclusion criteria. **b** Bar graph comparing the distribution of AHI symptoms between participants that were defined to be with and without ARS ($N = 33$). ARS was defined based on 11 AHI symptoms, and unobserved linkages between symptoms using Latent Class Analysis. Incremental latent group models were assessed to predict the goodness of fit. The model with two latent groups was the best fit, with the lowest BIC value (660.5) compared to three (678.6), four (699.2), or five (714.7) groups. Study participants were grouped based on their predicted posterior probabilities into those with ARS ($N = 20/33$ (60%)) and those without ARS (13/33 (40%)). **c** Box plots displaying the results of the cross-validated performance measure (accuracy) for the ARS PLS-DA models based on the following datasets: V0 + V1 + V2; V0 + V1-V0 + V2-V0; V1 + V2; V1-V0 + V2-V0; V1-V0; and V2-V0. The models were trained to predict ARS "Yes" or "No" and evaluated in 10 5-fold cross-validations, resulting in 50 individual accuracy values from 50 test sets. Each boxplot shows the distribution of accuracy values across 50 cross-validation models. The center line within the box represents the median, the box bounds the interquartile range, and the whiskers the minimum and maximum values of $1.5 \times$ IQR beyond the box. Any data points beyond these are considered outliers, and shown as individual points. **d** Score plot based on the V1-V0 + V2-V0 dataset (with the highest accuracy value) from (**c**), indicating the group membership of each sample. There was clear discrimination between the ARS-No (orange) and the ARS-Yes (green) samples on the first (x-axis) and second components (y-axis). Axis labels indicate the percentage of variation explained per component. **e** Boxplot showing the variable importance in projection (VIP) scores in the PLS-DA model based on V1-V0 or V2-V0 for each protein. The VIP score summarizes the contribution a variable (protein) makes to the model. This plot identifies the most important proteins for the classification of ARS "Yes" or "No". Proteins with high VIPs are more important in providing class separation. Black points represent the full model, and the boxplots indicate the distributions of 10 cross-validation models. The sample size corresponds to the 50 VIP scores computed for each protein across the 50 cross-validation models. The center line within the box represents the median, the box bounds the interquartile range, and the whiskers the minimum and maximum

values of $1.5 \times$ IQR beyond the box. Any data points beyond these are considered outliers, and shown as individual points. Red dots represent the VIP scores for the full PLS-DA model, capturing the importance of each protein feature in the model's ability to discriminate between groups. These points reflect the average or specific metric of the VIP values used to build the full model. **f** Forest plots indicating effect sizes (log2 fold change) and 95% confidence intervals for proteins significantly differentially expressed at 2 weeks and 1 month post estimated date of infection (EDI), relative to pre-infection levels (V1-V0 and V2-V0, respectively). Only individuals from the IAVI cohort were included since ARS data are only available for this cohort. Circles and triangles indicate depleted (depl) and neat plasma, respectively. The statistical analysis was conducted using linear mixed-effects models with a random intercept for each patient, treating visit number as a categorical variable. The differential protein expression was assessed using a global ANOVA, with post hoc tests identifying specific visit comparisons (e.g., V0 vs. V1, V0 vs. V2, and V1 vs V2). The Benjamini-Hochberg's FDR method with a 5% FDR threshold was used to correct for multiple testing, with a fixed $p$-value cut-off of 0.005. **g** Heatmap of proteins associated with ARS based on hierarchical clustering of the V1-V0 and V2-V0 expression of the selected proteins. The heatmap provides a visual representation of coordinated changes of the proteins identified through PLS-DA and linear regression in relation to ARS status. **h** Pirate plots showing the V1–V0 protein expression for the top proteins between those with and without ARS. **i** Table representing the longitudinal protein expression profiles for the top proteins associated with ARS. For each profile and protein, the number (%) of patients with or without ARS were recorded. Abbreviations: ARS acute retroviral syndrome, PLS-DA Partial Least Squares Discriminant Analysis, V0 visit 0 (collected before estimated date of infection), V1 visit 1 (collected 10–14 days post estimated date of infection), V2 visit 2 (collected 15–42 days before estimated date of infection), V1-V0 difference between visit V1 and V0, V2–V0 difference between visit V2 and V0, V2–V1 difference between visit V2 and V1, VIP variable importance in projection, Log2FC log 2-fold change. The asterisk (*) appended to the end of certain protein names indicates proteins detected in neat plasma, while proteins without an asterisk were identified in depleted plasma samples. Source data are provided as a Source Data file.

(at peak viremia) and were both persistently elevated after hAHI. vWF reflects persistent endothelial cell activation due to activation of the inflammation/coagulation pathway, whereas FN1 is involved in cell adhesion, cell motility, opsonization, wound healing, and maintenance of cell shape[43]. FN1 also binds to HIV-1 gp120, which has been shown to enhance complement interaction and infection of primary CD4 + T-cells[44]. Moreover, HIV-1 has been shown to modulate the host cytoskeleton dynamics[45]. For example, TTN, a protein that has been implicated in HIV-1 Gag subcellular trafficking, showed increased levels 1 month after infection, whereas FGL1 levels, a marker of T-cell activation/exhaustion, decreased at the onset of peak viremia, coinciding with the depletion of CD4 + T-cells 3–4 weeks post-infection[46,47]. Furthermore, HNRNPA2B1 levels at 4 weeks after infection have been associated with HIV-1 set-point levels, and previous studies have suggested that HNRNPA2B1 can interact with viral components, have a critical role in regulating the viral life cycle, and function as a viral DNA sensor initiating IFN production[48,49].

We have previously shown that a generally stronger innate immune response, and an IP-10 activation in particular, is associated with the manifestation of ARS[9]. In the current study, decreasing levels of ZYX, and SCGB1A1 during hAHI were associated with ARS. It is possible that decreased levels of SCGB1A1, a pulmonary surfactant protein that influences alveolar macrophage-mediated inflammation, may be linked to lung inflammation and epithelial integrity in participants with ARS[33,50,51]. Moreover, the associations between SCGB1A1 and ZYX expression and ARS supports previous suggestions of these proteins' involvement in regulating innate immune responses triggered by viruses[52,53]. Specifically, ZYX binds to the mitochondrial antiviral signaling protein following recognition of cytoplasmic double-stranded RNA by Retinoic acid-inducible gene I-like receptors. This latter interaction triggers the induction of type I interferon (IFN) expression,

which constitute the predominant immune response during hAHI. Like SGB1A1, LILRA3 plays a role in modulating inflammatory immune responses and it is possible that the decreased expression in participants without ARS reflects a dampened inflammatory response[34,54]. In addition, the absence of ARS in participants with these protein expression patterns suggest a more regulated immune response during hAHI, and proteins like Zyxin could serve as early indicators, allowing for prompt diagnosis and treatment initiation.

Previous studies have suggested that certain cytokines, such as interleukin (IL)-15, IL-7, IL-12p40, IL-12p70, and IFN-γ, can predict 66% of the variation in viral set-point 12 months after infection[55]. In this study, viral controllers had increased levels of NAPA and RAN during hAHI and decreased levels of ITIH4 in hAHI. Assembly and release of virus-like particles are promoted by a variety of host factors, for example RAN has been suggested to be a crucial protein involved in host interactions with the HIV-1 Rev protein[38]. More specifically, RAN facilitates the release of the Rev cargo, which is a prerequisite for binding of cargo proteins by nuclear export factors. Moreover, NAPA is involved in membrane fusion and vesicle trafficking, playing a critical role in intracellular transport. HIV-1 relies on host cellular machinery for viral assembly and budding, and it is possible that elevated expression of proteins like NAPA and RAN plays a role in maintaining the integrity of cellular processes that may inhibit viral release or replication in HIV-1 infected participants with lower viral load. Furthermore, ITIH4 is a plasma protein primarily involved in inflammatory responses and tissue repair, functioning as an acute-phase protein[56]. The lower ITIH4 expression in participants with low viral load may reflect a reduced systemic inflammatory state, which is often seen in individuals with better viral control[57].

The association between increased PRKCB expression and faster CD4 + T-cell decline may be related to PRKCB-induced Nuclear Factor-

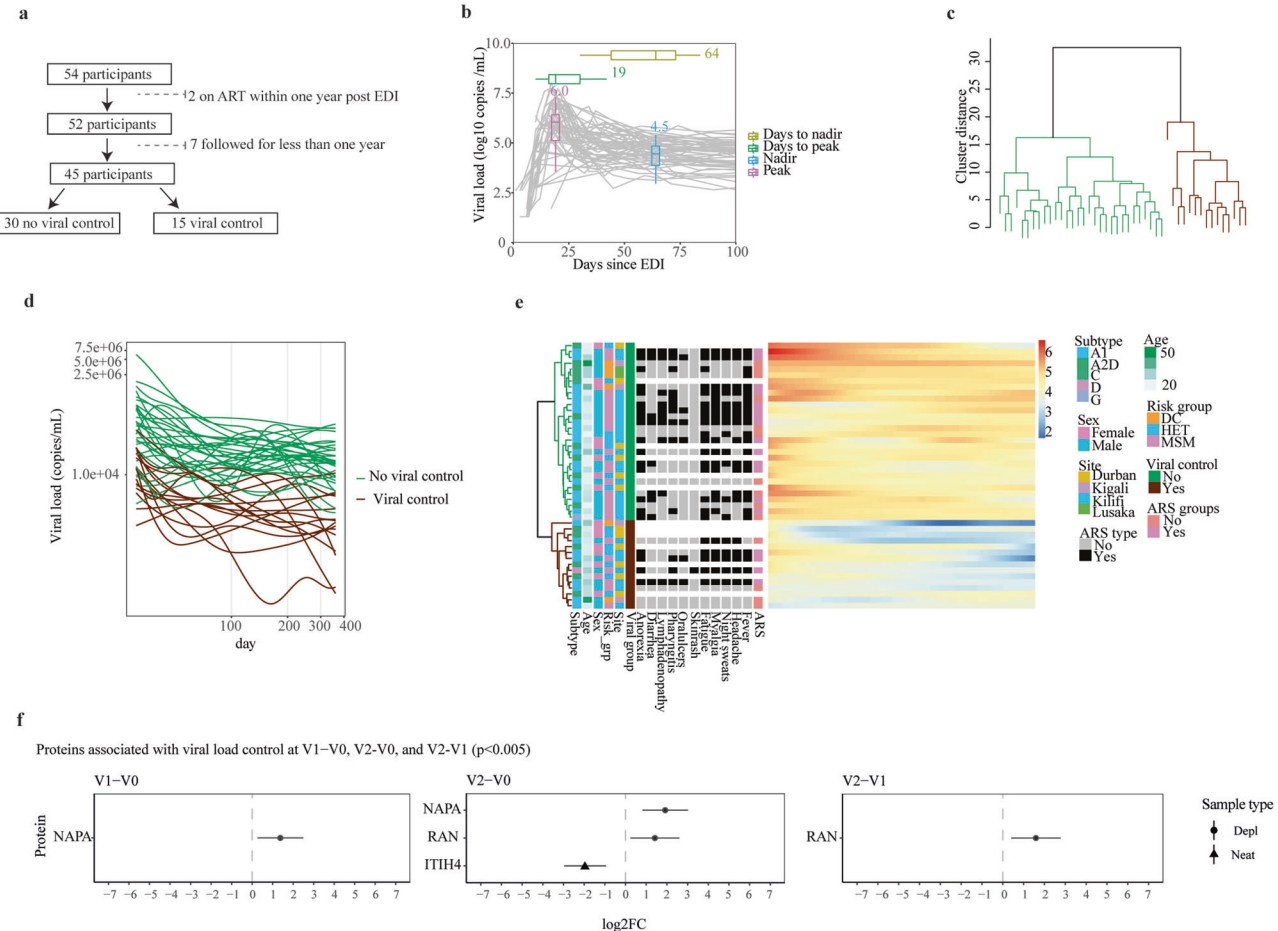

**Fig. 4 | Alpha-soluble NSF attachment protein, GTP-binding nuclear protein Ran, and Inter-alpha-trypsin inhibitor heavy chain are associated with HIV-1 control. a** Flow chart illustrating the total number of samples used in viral control classification, along with the exclusion criteria. **b** Longitudinal viral load measures against the number of days post the estimated date of infection (EDI) for all 54 participants. The color-coded boxplots represent the distribution of peak viral load, nadir viral load, days to peak viral load, and days to nadir viral load across the 54 individuals. The center line within the box represents the median, the box bounds the interquartile range, and the whiskers the minimum and maximum values of 1.5× IQR beyond the box. **c** Dendrogram showcasing complete linkage hierarchical clustering of viral load profiles. Euclidean distances computed from the cubic spline predicted viral load at evenly spread (on transformed scale) time points were used for clustering. The optimal number of clusters was determined using the Silhouette value, and the clustering significance was calculated using multiscale bootstrap resampling. Viral load clusters were based on time 1–12 months (30–364 days). Two distinct groups were classified: No viral control (in green) and sustained viral control (in brown). **d** Plot representing the cubic spline predicted viral load at evenly spread time points. The differentiation between the two viral control groups occurred at a viral load threshold of 10,000 copies/ml. **e** Heatmap illustrating associations between viral control and various demographic parameters and ARS symptoms. **f** Forest plots indicating effect sizes (log2 fold

change) and 95% confidence intervals for proteins significantly differentially expressed at 2 weeks and 1 month post estimated date of infection (EDI), relative to pre-infection levels (V1-V0 and V2-V0, respectively), as well as the difference between 2 weeks and 1 month post EDI (V2-V1). Circles and triangles indicate depleted (depl) and neat plasma, respectively. The statistical analysis was conducted using linear mixed-effects models with a random intercept for each patient, treating visit number as a categorical variable. The differential protein expression was assessed using a global ANOVA, with post hoc tests identifying specific visit comparisons (e.g., V0 vs. V1, V0 vs. V2, and V1 vs V2). The Benjamini-Hochberg's FDR method with a 5% FDR threshold was used to correct for multiple testing, with a fixed $p$-value cut-off of 0.005. Abbreviations: ART antiretroviral treatment, EDI estimated date of infection, ARS acute retroviral syndrome, DC discordant couple, HET heterosexual, MSM men who have sex with men, V0 visit 0 (collected before estimated date of infection), V1 visit 1 (collected 10–14 days post estimated date of infection), V2 visit 2 (collected 15–42 days before estimated date of infection), V1-V0 difference between visit V1 and V0, V2-V0 difference between visit V2 and V0, V2-V1 difference between visit V2 and V1, Log2FC log 2-fold change. The asterisk (*) appended to the end of certain protein names indicates proteins detected in neat plasma, while proteins without an asterisk were identified in depleted plasma samples. Source data are provided as a Source Data file.

kappa-B (NF-κB) activation, which regulates B-cell activation and binds to the HIV-1 promoter resulting in enhanced viral transcription. PRKCB is also involved in cytoskeletal rearrangements necessary for virus entry, further implicating its importance in HIV-1 replication and the potential exacerbation of CD4 + T-cell decline. Finally, increased levels of the host protease Hepsin were associated with a faster disease progression. This is in line with previous observations that Hepsin suppresses the induction of type I interferons (one of the major defense mechanisms of the human innate immune system towards virus infections)[58]. Understanding the proteomic differences between

fast and slow progressors can provide insights into the mechanisms driving HIV-1 disease progression. Ultimately, and if verified in targeted studies, these proteins have potential as future targets for therapeutic interventions.

The main strength of this study is the unique and well-characterized samples collected before, during and after hAHI. This enabled a large-scale assessment of how HIV-1 ARS, viral load and disease progression is associated with the plasma proteome dynamics in hyperacute HIV-1 infection at both population and individual levels relative to pre-infection levels. Still, this study is not without

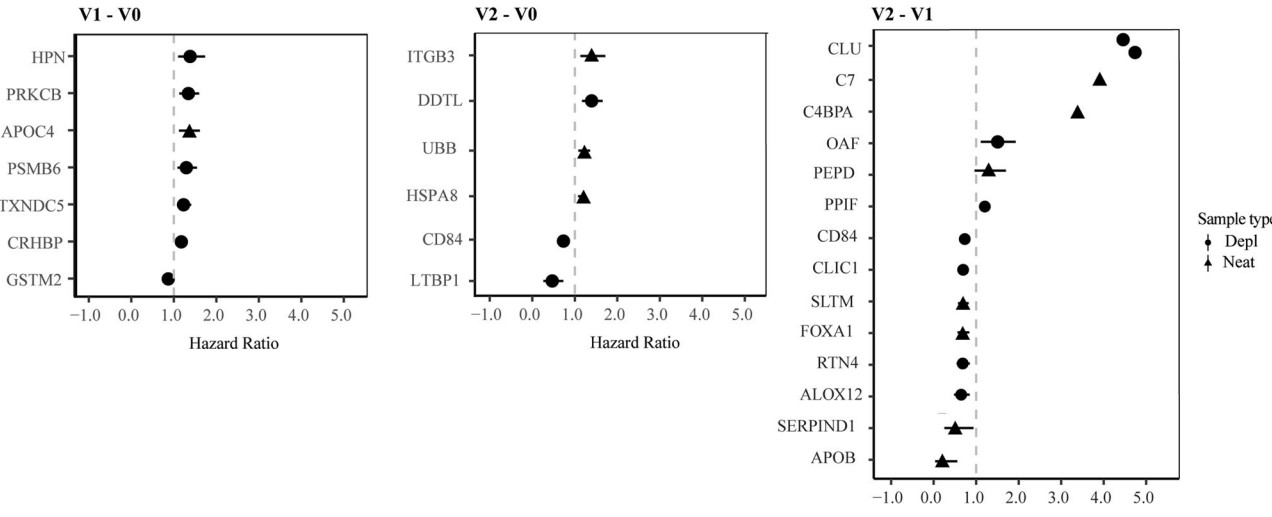

**Fig. 5 | Hepsin, Protein kinase C beta, and Integrin subunit beta 3 are associated with an increased risk of disease progression.** Forest plots indicating hazard ratios (HR) and 95% confidence intervals for proteins differentially expressed at 2 weeks and 1 month post estimated date of infection (EDI), relative to pre-infection levels (V1-V0 and V2-V0, respectively), as well as the difference between 2 weeks and 1 month post EDI (V2-V1). The Cox proportional hazards model was used to determine the association between plasma protein expression (independent variable) and the risk of disease progression. The event outcome was defined as CD4 + T-cell counts of 500 cells/μl from 6 weeks post the estimated date of infection. Covariates included age, sex, and cohort. Circles and triangles indicate depleted (depl) and neat plasma, respectively. Abbreviations: V0 visit 0 (collected before estimated date of infection), V1 visit 1 (collected 10–14 days post estimated date of infection), V2 visit 2 (collected 15–42 days before estimated date of infection), V1–V0 difference between visit V1 and V0, V2–V0 difference between visit V2 and V0, V2–V1 difference between visit V2 and V1. Source data are provided as a Source Data file.

limitations. First, platelet contamination during plasma separation has been proposed as a potential bias in proteomic studies[59]. Although consistent protocols were employed between cohort sites, the possibility of human error affecting sample collection and handling cannot be fully excluded. Second, cohort differences, such as genetic, demographic (e.g. the study population was skewed towards MSM), geographic, and pathogenic burden between IAVI and Durban participants, may have influenced protein expression patterns between cohorts, as well as the generalizability of our findings. Third, technical variability in sample processing, measurement techniques and batch effects in proteomics analysis can introduce bias, despite our efforts to normalize and control for such factors. It is also possible that the depletion process inadvertently removed some untargeted proteins. Still, the overall quantification of proteins increased threefold by this approach, and importantly, the majority of proteins that were detected in both the neat and depleted approaches showed similar results. Future studies with larger, more diverse cohorts and additional time points are needed to validate and further explore the findings in this study.

In summary, we provide insights into plasma protein dynamics associated with the complex virus-host interactions and responses that take place when HIV-1 establishes infection in the human host, and highlight a similar role of both protein *stormers* and *slumpers* during hAHI. Several predictive and prognostic biomarkers associated with ARS, VL responses, and disease progression were identified. The potential implications of these findings are substantial and pave the way for future investigations of diagnostic and prognostic utilities of these biomarkers. Furthermore, our study underscores the need for continued basic research of HIV-1 infection, and other virus infections, to identify biomarkers with potential for both early diagnosis, and improved treatment strategies of viruses.

## Methods
### Study participants and ethical considerations
This study includes data and samples from sub-Saharan African (sSA) adults (≥18 years old), recruited in two distinct acute and early HIV-1 infection cohorts: The IAVI cohort and Durban cohort. The IAVI cohort comprises African participants enrolled in Kenya, Rwanda, and Zambia between 2006 and 2011 under IAVI's protocols B and C, while Durban cohort consists of South African participants enrolled in Durban between 2007 and 2014 under the FRESH (Females Rising through Education Support and Health) and HIV Pathogenesis Programme acute infection cohorts[23–26]. Baseline variables such as date of birth, sex, HIV-1 RNA, HIV-1 p24 antigen, and antibody test results, date of HIV-1 diagnosis, transmission risk group, antiretroviral treatment start date, CD4+ and CD8 + T-cell dynamics were collected from both cohorts. ARS data was only available for the IAVI cohort.

All study participants provided written informed consent for the use of their samples for biomedical research. All sites received approvals from respective country-specific ethics review boards. For the IAVI cohort: Kenya Medical Research Institute Ethical Review Committee; the Kenyatta National Hospital Ethical Review Committee of the University of Nairobi; the Rwanda National Ethics Committee, the Uganda Virus Research Institute Science and Ethics Committee; the Uganda National Council of Science and Technology; the University of Zambia Research Ethics Committee; and the Emory University Institutional Review Board[23,24]. For the Durban cohort: The University of Cape Town Health Science Research and Ethics Committee; the Bio-Medical Research Ethics Committee at the University of KwaZulu Natal; and the institutional review board of Massachusetts General Hospital[25,26]. Compensation was not provided to patients for participation in this study. All data and samples were de-identified and anonymised to protect the privacy of the participants.

Eligibility for the study included hyperacute HIV-1 infection (hAHI), defined as HIV-1 antibody negative and RNA positive (Fiebig stage I), or p24 antigen-positive (Fiebig stage II), corresponding to the period from onset of plasma viremia to peak viral load[5–8]. Estimated date of infection (EDI) was defined either as the midpoint between the date of the last negative and first positive HIV antibody test, 14 days before the date of the first positive p24 antigen test (with a negative antibody test), or 10 days before the date of the first PCR-positive test (with a negative antibody or p24 antigen detection). Matched

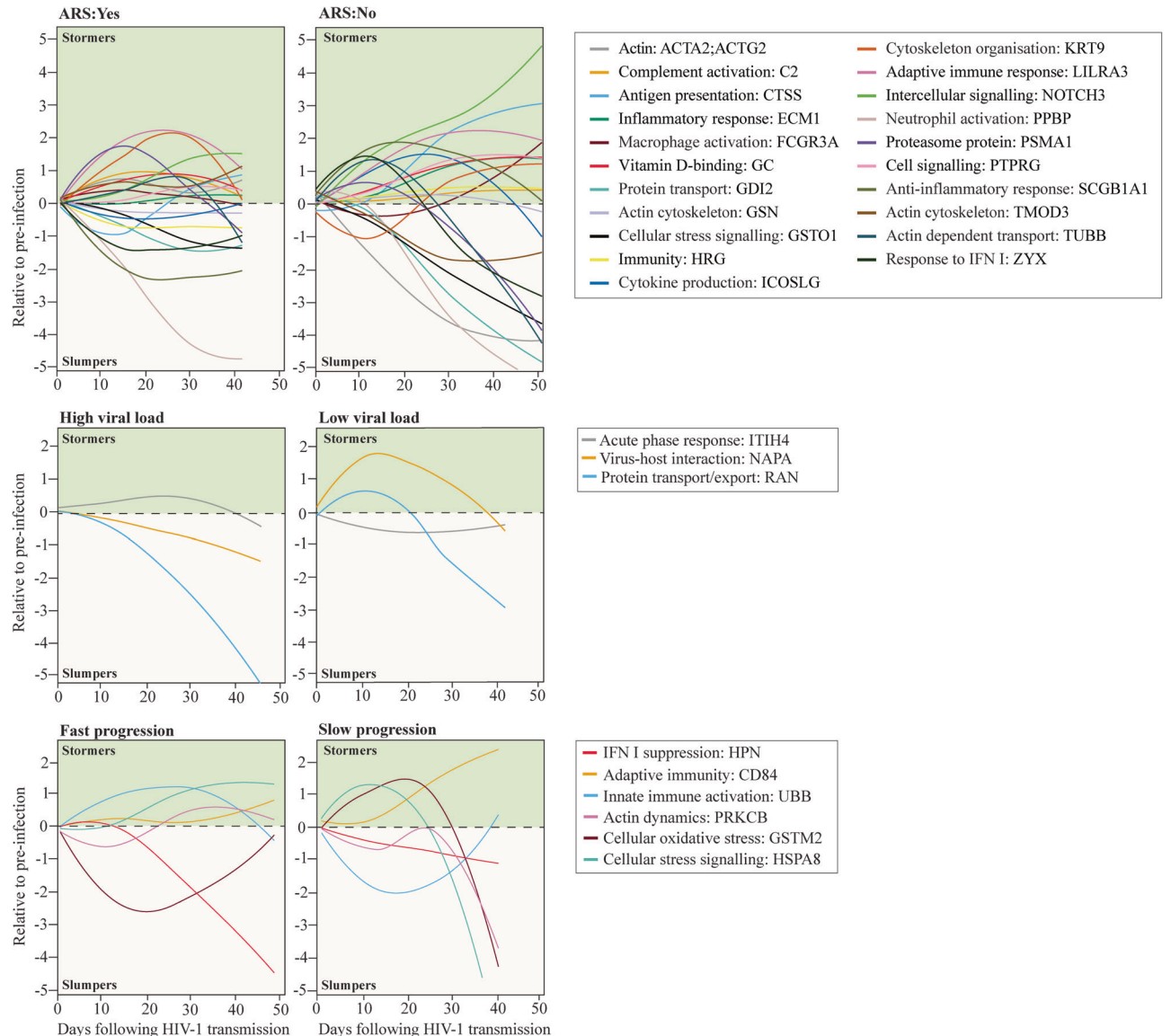

**Fig. 6 | Longitudinal dynamics during hyperacute HIV-1 infection of key differentially expressed proteins associated with ARS, viral load, and disease progression.** This schematic illustrates the temporal changes in expression for proteins associated with ARS, viral load, and disease progression during acute phase of HIV-1 infection. Protein expression levels were assessed both before infection, and within the two to 6 weeks following infection in the 54 study participants. Participants were categorized into subgroups based on various outcomes, including ARS (presence or absence), viral control status (controllers or non-controllers), and the rate of disease progression (fast or slow). The x-axis of the graph

represents the time in days following infection when plasma samples were collected, while the y-axis represents the mean protein expression levels relative to the pre-infection baseline. Key proteins associated with ARS, viral control and disease progression are depicted using smoothed lines generated through local regression plotting with a span of 1.5. The selected proteins represented the top proteins associated with ARS, viral load, and disease progression. To enhance clarity and highlight distinctive patterns, proteins with similar dynamics in the group comparisons were excluded. Abbreviations: ARS acute retroviral syndrome.

longitudinal plasma samples were collected at three different visits: (i) visit 0 (V0), 22 and 120 days before EDI; (ii) visit 1 (V1), 10–14 days post EDI; and (iii) visit 2 (V2), 15–42 days post EDI.

**Sample preparation for LC-MS/MS analysis**
One-hundred and fifty-seven blood plasma samples archived at −80 °C were obtained for liquid chromatography with tandem mass spectrometry (LC-MS/MS). Each sample was analyzed both neat and depleted to increase the number of detected proteins. Depletion refers to the removal of the top 14 highly abundant proteins before protein quantification. Due to system migration, neat plasma samples from the Durban cohort were processed by Q Exactive HF-X mass spectrometer, and samples from the IAVI cohort were processed by Exploris 480

mass spectrometer with FAIMS. Importantly, this did not affect the number/intensity of proteins quantitated. Details of the specific plasma preparation, LC-MS/MS run conditions, instrumentation, and spectral library are presented in Supplementary methods.

**DIA/SWATH-MS targeted data extraction**
DIA data files were analyzed against the spectral library using the BGS factory default settings in Spectronaut 15.1 (Biognosys, Schlieren, Switzerland). The identifications were filtered at a false discovery rate (FDR) of 1% at both peptide and protein levels. Spectronaut used retention time prediction based on iRT, the m/z dimension in the SWATH-MS data, mass accuracy, and isotopic distribution of fragment ions to identify peptides. All available transitions were extracted for

each targeted peptide together with their corresponding decoy-transition groups generated by pseudo-reversing the sequence of the targeted peptides.

## Protein quantification and data pre-processing

To derive protein abundances, peptide precursor and fragment ion intensities were used together with the MaxLFQ algorithm (implemented in *iq* R Package). This algorithm combined multiple peptide ratios to derive optimal protein ratios between pairs of samples, ensuring the accuracy and reliability of the data[60,61]. The global distribution of protein signals was assessed to identify poor-quality or low-intensity data. Proteins with >80% missing values across samples were removed, and factors such as time of sample collection, date of infection, and date of MS data acquisition were assessed to identify any correlations with missingness. Missing values were imputed by replacing them with a randomly chosen value between one and the minimum global raw intensity value of the protein. The data matrix was then analyzed by NormalyzerDE to identify the normalization method with the least variance in the data, and the normalization was performed based on the results obtained from NormalyzerDE[62].

## Data analysis

**Differential expression analysis.** To identify factors contributing to technical variation in the data, Principal Component Analysis was used. This analysis was conducted to test the impact of variables such as site of collection, HIV-1 subtype, age, date of processing, and visit number. To account for observed variation, adjustments for principal components (PC1 and PC2) was done by the inclusion of the PCs in mixed effects models. Linear mixed models with a random intercept for each study participant at all time points and an interaction term between visit and cohort were used to identify differentially expressed proteins (DEPs) over time. This method is designed to handle large sample sizes, and effectively accounts for continuous distributions of quantitative response variables, and thereby reduces inflation of type I error rates[63]. In the models, the dependent variable was the normalized log2 protein intensities for each time point, with visit as fixed effect. Age, sex, cohort, and HIV-1 subtype were assessed a priori as potential confounders for all subsequent analyses based on existing literature and biological relevance. Model parameters are summarized in Table S9. A global analysis of variance (ANOVA) was then conducted to identify proteins that changed between visits, and post hoc tests were performed to determine when the changes occurred. Global *p*-values were determined for each protein by likelihood ratio tests of the full model with the effect in question against the model without the effect. The Benjamini-Hochberg's FDR method with a significance threshold of 5% FDR was used to correct for multiple testing. In addition, a fixed *p*-value cut-off of $p < 0.005$ for all tests was applied. The up and downregulated DEPs at different visits were filtered using $p < 0.005$, and plotted in Volcano or Forest plots with 95% CI for log2 fold change.

**Enrichment or pathway analysis.** Protein classifications and annotations were based on subcellular location annotations from the uniProt database, which denotes location and the topology of the mature protein in the cell, and the human protein atlas[22,30]. To determine whether a set of differentially expressed proteins and their gene ontology (GO) biological processes were statistically different between two biological states (activated and suppressed), Clusterprofiler was used to perform an enrichment DE analysis[64]. The statistical significance of overrepresentation was determined by Fisher's exact test and Bonferroni's FDR ($p < 0.05$). The protein list from both neat and depleted plasma was used as custom background. The top ten enriched terms, along with their respective $p$ values, were determined.

**Tissue damage analysis.** To evaluate tissue-specific protein expression, a previously established dataset of tissue-enriched

transcriptional signatures derived from the Genotype-Tissue Expression (GTEx) project was utilized[31]. The GTEx read counts were transformed into trimmed values and normalized to z-scores for each gene across tissues. Genes with a z-score exceeding three were classified as tissue-enriched. Subsequently, the list of tissue-enriched proteins was used as the reference database for an enrichment analysis with all quantified proteins using Clusterprofiler.

**Acute Retroviral Syndrome (ARS).** Symptoms during acute HIV infection (AHI) were recorded using a standardized questionnaire 2–6 weeks after the estimated date of infection (only for the IAVI cohort since this was not part of the study protocol for the Durban cohort)[11]. ARS was defined based on the 11 symptoms fever, headache, myalgia, fatigue, anorexia, pharyngitis, diarrhea, night sweats, skin rash, lymphadenopathy, and oral ulcers. Previous studies have used different definitions for ARS, such as reporting any symptom, ≥2 symptoms, ≥3 symptoms, or a combination of fever with other symptom(s)[65]. However, discrete classification methods may not account for unobserved linkages between symptoms. To address this limitation, latent class analysis (LCA), a structural equation modeling approach, was used to group participants based on the number of AHI symptoms and other unobserved linkages, as previously described[9]. Partial Least-Squares Discriminant Analysis (PLS-DA) was used to simultaneously identify a group of proteins associated with ARS. This supervised dimensionality reduction method incorporates class labels to find the direction of maximum class separation, making it well-suited for classification (here, participants with and without ARS). The response variable was the binary ARS status (presence or absence of ARS). Predictor variables included the normalized expression levels of proteins measured in plasma samples. A combination of cross-validation and permutation testing to assess the models predictive power and robustness. Moreover, the classification accuracy, sensitivity, specificity, and the area under the receiver operating characteristic (ROC) curve was used to assess the model's discriminatory ability. Variable importance in projection (VIP) scores were used to identify the most influential proteins contributing to the discrimination between ARS-positive and ARS-negative participants. Proteins with VIP scores >2 were considered significant contributors.

Following the identification of candidate proteins through PLS-DA, linear mixed-effects models were used to evaluate the association of each protein individually with ARS. Specifically, the model assessed how ARS, time (visit), age, and their interaction (ARS*time) influence logI, while also accounting for differences in baseline logI between participants through the random intercept. Model parameters are summarized in Table S9. This step provided detailed insights into how the expression levels of specific proteins relate to ARS. Only participants with protein values from V0, V1, and V2 were included in the analysis, with age as a covariate.

**Disease progression.** Disease progression was measured using two endpoints: Viral control and CD4 + T-cell decline. For viral control, viral load measurements were taken on various days for each study participant. Curve fittings were used to compare viral load profiles between participants. All VL measurements from EDI to the start of antiretroviral treatment (ART) were used. VL measurements were log10-transformed, and an optimal smoothing parameter was calculated using leave-one-out cross-validation. A cubic smoothing spline was then fitted separately for each participant, and the Euclidean distance between VL curves was calculated at evenly distributed time points. Participants with observations at the beginning and end of a given time interval were clustered based on Euclidean distance using complete linkage hierarchical clustering. The optimal number of clusters was determined using the Silhouette method, and clusters were based on a period of 1–12 months. Fisher's exact, Chi-square, and Mann-Whitney $U$ tests were performed to assess the association between clinical

parameters and VL clusters or disease progression group classifications. Linear mixed-effect models were used to assess how viral control, time (visit), age, cohort and the interaction term viral control*time influence logI, while also accounting for differences in baseline logI between participants through the random intercept. Model parameters are summarized in Table S9. In addition, Fisher's exact tests were performed to assess the association between HLA types and VL clusters or disease progression group classifications.

For CD4 + T-cell decline, a time-to-event analysis was conducted using an absolute CD4 + T-cell count <500 from 6 weeks after EDI as the event. The log-rank test was used to evaluate differences in time to event between cohorts ($p < 0.05$ was considered statistically significant). For clarity, the Cox proportional hazards model was used to determine the association between plasma protein expression (independent variable) at each visit or visit difference and the risk of disease progression. Follow-up time was censored at the initiation of ART, or the last observed time point (if ART was not initiated during the study time). The results were presented as Hazard ratios with 95% confidence intervals, and Kaplan-Meier time-to-event curves.

### Reporting summary
Further information on research design is available in the Nature Portfolio Reporting Summary linked to this article.

## Data availability
The proteomics data generated in this study have been deposited in the ProteomeXchange Consortium via PRoteomics IDEntifications (PRIDE) partner repository with the dataset identifier PXD042850. Source data are provided as a Source Data file. All additional data are available within the article, Supplementary Files. Source data are provided with this paper.

## Code availability
All original code has been deposited and can be assessed at: https://github.com/jnazziwa/AHI_Plasma_Proteomics.

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

## Acknowledgements

The authors thank IAVI for supporting HIV-1 research studies and capacity-building initiatives in Kenya, Rwanda, and Zambia. They are also grateful to staff and participants from IAVI's protocol B and C sites in Africa, without whom this work would not have been possible. They also acknowledge the Swedish National Infrastructure for Biological Mass Spectrometry (BioMS), and the following people for their contributions, and support: Malin Neptin (Department of Translational Medicine, Lund University, Sweden), Ashfaq Ali (NBIS expert), Jakob Wilforss (Department of Immunotechnology, Lund University, Sweden), Christofer Karlsson (Division of Infection Medicine, Department of Clinical Sciences Lund, Faculty of Medicine, Lund University, Sweden), Hong Yan (Department of Clinical Sciences, BioMS, Lund University, Sweden), and Johan Malmström (Division of Infection Medicine, Department of Clinical Sciences Lund, Faculty of Medicine, Lund University, Sweden). This work was supported by the generous support of the American people through the United States Agency for International Development (USAID). The contents are the responsibility of the study authors and do not necessarily reflect the views of USAID, the National Institutes of Health (NIH), or the US government. This work was also supported by funding from the Swedish Research Council (grant numbers 2016-01417 and 2020-06262 to J.E.) and the Swedish Society for Medical Research (grant number SA-2016 to J.E.). The authors are also grateful for the support of the Sub-Saharan African Network for TB/HIV-1 Research Excellence (SANTHE), a DELTAS Africa

Initiative (grant number DEL-15–006 to A.S.H.). with support by the Wellcome Trust (grant number 107752/Z/15/Z), the UK Foreign, Commonwealth & Development Office, through the Developing Excellence in Leadership, Training and Science in Africa (DELTAS Africa) program. J.N. was funded by the Swedish Research Council (grant numbers 2016-01417 and 2020-06262) and the Medical Faculty at Lund University. A.S.H. was supported by a training fellowship from the Wellcome Trust (209294/Z/17/Z).

## Author contributions

J.E. conceived the study. J.E. and A.S.H. supervised the research. J.E, A.S.H., and J.N designed the study. J.N. and J.E. coordinated experiments and wrote the manuscript with inputs from all authors. J.N., K.G., and A.S.H. curated the data. J.N., M.R., T.M., and S.K. performed the sample preparation, LC-MS/MS analysis, and experiment optimization. J.N., E.J., E.F., M.H., and F.A. wrote and optimized the original code used in the study and performed the data analysis. J.N., E.J., E.F., M.H., F.A., A.S.H., and J.E. interpreted the results. All authors reviewed and approved the final version of the manuscript. The following authors managed the sites to which participants were attended and where samples were collected, provided clinical expertise and/or provided clinical data for the participant samples: J.H., A.K., E.K., W.K., M.A.P., P.K., S.A., E.H., T.N., J.G., S.R.J., and E.J.S.

## Funding

## Competing interests

The authors declare no competing interests.

## Additional information

[1]Department of Translational Medicine, Lund University, Lund, Sweden. [2]Lund University Virus Centre, Lund University, Lund, Sweden. [3]National Bioinformatics Infrastructure Sweden, Science for Life Laboratory, Department of Cell and Molecular Biology, Uppsala University, Uppsala, Sweden. [4]National Bioinformatics Infrastructure Sweden, Science for Life Laboratory, Department of Biochemistry and Biophysics, Stockholm University, Stockholm, Sweden. [5]BioMS–Swedish National Infrastructure for Biological Mass Spectrometry, Lund University, Lund, Sweden. [6]IAVI Human Immunology Laboratory, Imperial College, London, UK. [7]IAVI, New York, NY, USA. [8]IAVI, Nairobi, Kenya. [9]Africa Health Research Institute, Durban, South Africa. [10]HIV Pathogenesis Programme, The Doris Duke Medical Research Institute, University of KwaZulu-Natal, Durban, South Africa. [11]Division of Infection and Immunity, University College London, London, UK. [12]Department of Biomedical Engineering, Faculty of Engineering, Lund University, Lund, Sweden. [13]Division of Infection Medicine, Department of Clinical Sciences Lund, Faculty of Medicine, Lund University, Lund, Sweden. [14]Center for Family Health Research, Kigali, Rwanda. [15]Center for Family Health Research, Lusaka, Zambia. [16]UCSF Department of Epidemiology and Biostatistics, San Francisco, CA, USA. [17]Uganda Research Unit, Medical Research Council/Uganda Virus Research Institute and London School of Hygiene and Tropical Medicine, Entebbe, Uganda. [18]Department of Pathology & Laboratory Medicine, School of Medicine, Emory University, Atlanta, GA, USA. [19]Ragon Institute of Massachusetts General Hospital, Massachusetts Institute of Technology and Harvard University, Cambridge, MA, USA. [20]Department of Infectious Diseases, Infection and Immunity, Faculty of Medicine, Imperial College, London, UK. [21]Nuffield Department of Medicine, University of Oxford, Oxford, UK. [22]Sir William Dunn School of Pathology, University of Oxford, Oxford, UK. [23]KEMRI/Wellcome Trust Research Programme, Kilifi, Kenya. [24]The Aurum Institute, Johannesburg, South Africa. [25]Institute for Human Development, Aga Khan University, Nairobi, Kenya. [26]These authors contributed equally: Amin S. Hassan, Joakim Esbjörnsson. ✉e-mail: joakim.esbjornsson@med.lu.se

