## [Transparent Peer Review file · Nature Communications]

Dynamics of the blood plasma proteome during hyperacute HIV-1 infection

Corresponding Author: Professor Joakim Esbjornsson

Version 0:

Reviewer comments:

Reviewer #1

(Remarks to the Author)

I would like to commend the authors for this work, which investigated the dynamics of the plasma proteome during hyperacute HIV-1 infection.

The main strength of the study is the longitudinal analysis of the proteome, albeit from a small rather selective/non-representative population (mostly young men, and identifying as having sex with men). The broader applicability of the results should therefore be approached with caution, although the research remains important.

The laboratory methods for the analysis of plasma proteome are sound and described with sufficient in detail. The data analysis is however missing key information to replicate the analysis. Although the codes were provided, it is very difficult to navigate through the uploaded folders, and finding what exactly you are looking for is not an easy task.

Below are my comments on the described data analyses:

PCA showed clustering of cohort and visit, and was used as a basis for confounder selection. Confounders should be selected a-priori and should include all necessary factors regardless of whether the PCA show clustering on these variables or not. Simply stated, PCA is not a tool for confounder selection. It only tells us the a big part of the variability in the data is contributed by cohort and visit. This does not indicate whether a certain variable would influence the relationship between exposure and outcome.

It is also intriguing that the set of confounders differ a lot depending on the analysis. Some are controlled for cohort only (timepoint differences), age only (association with ARS), age + cohort (HIV control), and age+sex+cohort+subtype (Cox regression). It is my opinion that all these (age, cohort, sex and subtype) are potential confounders in all these analyses and should be accounted for. The authors may also consider cohort as a random effect to allow for each cohort to have a different slope and intercept. This may make the analysis better as genetic and environment factors may be very different in the 2 countries.

PLS-DA was used for the multivariate analysis of ARS with an average accuracy of 78% assessed through cross-validation. PLS-DA is very sensitive to overfitting, especially in small sample sizes, hence more model parameters are needed to demonstrate that the model is robust. The authors are encouraged to also show the AUROC, misclassification error (confusion matrix shown), and most importantly an external validation (which given the small sample size might not be possible).

The authors also mentioned that a linear regression model was performed to CONFIRM the association between the identified proteins in the PLS-DA and ARS. I would like to remind the authors that univariable analysis and multivariable analysis serve different purposes and do not confirm each other. PLS-DA should be viewed entirely as a collective contribution of proteins groups. Therefore, proteins that score high in PLS-DA may do so because they are together with other proteins despite not having a substantial effect individually.

After doing PLS-DA and linear regression, the authors follow with a hierarchical clustering to identify the top proteins that distinguished participants with or without ARS. This is a lot of steps to get to the point. It is unclear to me why these steps

were taken when the PLS-DA and linear regression can already provide top hits of proteins.

Considering the time scale of the analyses, and the case-control type design of the study (ARS vs no-ARS, HIV controller vs non-controller), the authors are encouraged to analyse the data using logistic regression instead of linear regression, which I believe to be a more appropriate tool for this analysis since the proteins are measured before the outcome was observed, and the interest is to determine which proteins are associated with the development of the outcomes. These analyses should be adjusted for the appropriate confounders (including age, sex, cohort at the minimum).

Regarding the Cox regression, the interpretability of the hazard ratios is questionable. Since the protein levels are differences, what do these HRs really mean? A 1 point increase in the difference between V1 and V0 for PRKCB is associated with a 30% in risk for getting CD4+ below 500?? This seems difficult to believe. If the data is normalized and/scaled, then the interpretation is ever more tricky.

This question relates to one of the main pitfalls when using differences (or change scores) as predictors (or outcomes) in linear regression – interpreting the result beyond saying that the result is “significant”.

Another issue is regression to the mean. Taking the difference between two variables assumes that all other conditions are the same during those two time points, which is hardly true. There could be external factors (change in diet, infections, for example) that influence the variability in V0 that are no longer there during V1 (vice versa). Hence, V1 – V0 may be heavily confounded. Moreover, when using difference scores, regression to the mean can occur if extreme values at one time point tend to be closer to the mean at the subsequent time point. This can lead to an apparent change in the outcome variable that is not related to disease progression.

To mitigate these, the authors are encouraged to also consider performing the analysis with V1 or V2 as the outcome, and adjusting for V0 instead of regressing on the differences. This way, the HRs are more interpretable and we reduce the other problems that are described above.

Another solution could be to use the latent clusters as the exposure instead of the protein levels themselves. The authors have already determined which proteins move in the same way. It might help to say that the hazard of disease progression is higher among those that follow pattern A compared to pattern B, and by an HR of this much.

Final remarks, the labels in Figure 3 c and e do not make sense. What do V0V1V2, V0V10V20 and others mean?

In summary, I am convinced that this is an important topic. However, I am keen to see an improved data analysis before I can be fully convinced of the results.

Reviewer #2

(Remarks to the Author)

Jamirah Nazziwa et al., reported a comprehensive longitudinal study of proteomes (DIA-MS) from precious retrospective plasma samples collected before, during, and after hAHI in 57 HIV infected participants from four sub-Saharan African countries. The in silico analysis of the proteome results have been associated with different outcomes including: i) ARS in hyperacute HIV-1 infection, ii) Virus control and iii) risk of disease progression. Then a longitudinal dynamic of key factors has been also evaluated. From this analysis different candidates have emerged, such us: Zyxin (ZYG), Secretoglobin family 1A member 1 (SCGB1A1), and Pro-platelet basic protein (PPBP) for ARS; Rho GTPase activating protein 18 (RHGAP18), Annexin A1 (ANXA1), and Lipopolysaccharide binding protein (LBP) for viral loads; and Hepsin (HPN), Protein kinase C beta (PRKCB), and Integrin subunit beta 3 (ITGB3) for disease progression.

1) Given the proximity of V0 to seroconversion, I was wondering if you have assessed reservoir levels at peripheral blood. In this regard, do you have such information for later points?

2) Durban vs IAVI cohorts display market differences...age, gender, subtype, depletion vs non protein depletion, instrument DIA/MS...and it is not clearly explained the control for cofounders and co-variables in each analysis performed. In that sense, aside from the many sources of variability (differences in sex, age, subtype, African region, protein depletion vs non-depletion, instrument etc...) different models have been applied PLS-DA, Mixed linear model (Age, cohort), COX model (age, sex, cohort, substudy) being very confusing the rationale and the argument behind the selection of the model and the variable controls in each of them as well as samples used. It is recommended to detail this information in data analysis section in methods.

3) HIV controllers and rapid/slow progressors information: these participants have been HLA typed ? There is information as well for CCR5 delta 32?

4) Many genes are mentioned in the text in abbreviated form (gene ID) without first mentioning their full names.

5) There is a lack of study limitations in the discussion section considering all the variability sources and how the gender, age, African region, subtype information, technical performance that needs to be more explained.

6) The results obtained in this study are detailed descriptively, and although there is a literature search for the molecules

found to be relevant in this study and they are framed in the context of HIV, the findings are not discussed in depth in the clinical context outlined in the study's objectives (associations with ARS, viral load responses, and HIV-1 disease progression) and the impact they have or could have as blood biomarkers and/or targets for prophylactic and therapeutic HIV-1 interventions.

7) An attempt has been made to evaluate the relationship between identified molecules (correlations, ontologies, co-expression, gene neighborhood, protein homology...)? In fact, it would be interesting to see how those molecules that depict the same longitudinal dynamics during hyperacute HIV-1 infection (stormers and slumpers), even if their outcome relationship is different (ARS, viral load, or disease progression), could be related. This way, the molecules would not be treated independently but rather holistically as intended.

Version 1:

Reviewer comments:

Reviewer #1

(Remarks to the Author)

The authors argued that their population is not dominated by men having sex with men (MSM). According to the response, "the study population was somewhat skewed towards men (34 of 54 study participants), but less so towards MSM (28 of 54 study participants)."

If 34/54 are men, and 28/54 are MSM, this means that 28/34 of the men are MSM (82%). Out of 54 participants, 52% are MSM (28/54), 11% men not MSM (6/54), and 37% are women (20/54). I hope the authors can now clearly see and agree that the population is skewed towards MSM.

I am delighted that the authors found the suggested revisions in the data analysis useful in improving the manuscript. The authors did a great job in revising the data analysis, which is now more convincing.

I only want to inform the authors that the use of change scores is an interesting debate in the statistics literature. Several papers* have demonstrated that using change scores is not appropriate for linear regression models, and instead using V1 as outcome, adjusting for V0 is a more appropriate approach. The explanation of the authors: "However, using V1 or V2 as the outcome, and adjusting for V0 does not directly respond to the question whether protein value changes between visits are related to ARS. Instead, it responds to whether a difference in V1 protein levels between ARS Yes and No are independent of protein levels at V0." is a common misunderstanding of the linear model. In fact, the way to look at this is that "what is the difference in the levels of V1 between ARS if both ARS conditions have the same value at V0 (holding V0 constant)." Looking at it this way, one can see that any change in V1 (the V1 estimate), while holding V0 constant, actually reflects the change between V1 and V0.

*Interesting reads:

1. Peter W G Tennant, Kellyn F Arnold, George T H Ellison, Mark S Gilthorpe. Analyses of 'change scores' do not estimate causal effects in observational data. *International Journal of Epidemiology*, Volume 51, Issue 5, October 2022, Pages 1604-1615, <https://doi.org/10.1093/ije/dyab050>
2. Mattes, A., Roheger, M. Nothing wrong about change: the adequate choice of the dependent variable and design in prediction of cognitive training success. *BMC Med Res Methodol* 20, 296 (2020). <https://doi.org/10.1186/s12874-020-01176-8>

-> Paper 2 says that change scores are ok as long as the linear model is also adjusted for baseline. So, $\Delta(V1-V0) \sim ARS + V0$

However, I agree that since this remains a debated topic (or even an unknown issue) in the biological sciences field, my comment should not hinder publication of this important work. I hope the authors can have a deeper look at this for their future work.

The opinion of the editor is sought with regards to the use of hierarchical clustering. I understand the point of the authors that they want to serve the non-statistically inclined reader an interesting visual of the data. However, for readers that are proficient with data analysis, this approach may cause confusion. Sometimes less is more. People who read *Nature Communications* tend to have higher order thinking skills. We probably do not need to pamper them too much by giving them data renderings that are not needed.

In conclusion, I highly appreciate the effort made by the authors in revising the manuscript. Depending on the opinion of the editor regarding the hierarchical clustering, I agree that this important piece of scientific research should be published. Congratulations from this humble reviewer.

Extremely minor comment:

Line 2 introduction: I think it is better to keep the word "individuals" instead of "participants". The sentence is a general statement about humans and not necessarily those involved in the study.

(Remarks on code availability)

Reviewer #2

(Remarks to the Author)

All the questions and comments has been satisfactorily answered.

(Remarks on code availability)

Comments from Reviewer #1

Comments to the Author

I would like to commend the authors for this work, which investigated the dynamics of the plasma proteome during hyperacute HIV-1 infection.

The main strength of the study is the longitudinal analysis of the proteome, albeit from a small rather selective/non-representative population (mostly young men, and identifying as having sex with men). The broader applicability of the results should therefore be approached with caution, although the research remains important.

Response: We sincerely thank Reviewer #1 for commending the importance and uniqueness of our study, and also for pointing out some of the limitations. As described in the first paragraph of the Results section, the study population was somewhat skewed towards men (34 of 54 study participants), but less so towards MSM (28 of 54 study participants). In compliance with Reviewer #1's comments, we have now added a more detailed limitations section in the revised manuscript, including the need for caution in generalizing of the results (p. 18-19, lines 15-25; 1-3).

The laboratory methods for the analysis of plasma proteome are sound and described with sufficient in detail. The data analysis is however missing key information to replicate the analysis. Although the codes were provided, it is very difficult to navigate through the uploaded folders, and finding what exactly you are looking for is not an easy task.

Response: We thank Reviewer #1 for the important and thorough review of the information needed for our peers to replicate the analysis. To enhance the reproducibility and ease of navigation through our data analysis, we have made the following improvements:

- (1) *Improved documentation:* This includes adding README files to each folder, detailing the purpose of the folder, the contents, and instructions on how to navigate and use the files. We have added code annotations explaining the steps and the rationale behind each analysis. Furthermore, each dataset is accompanied by a metadata file describing the variables, units of measurement, and any preprocessing steps applied
- (2) *Simplified folder structure:* We have reorganized the folder structure to follow a more logical hierarchy, making it easier to locate specific files. File and folder names have been standardized and made more descriptive to facilitate quick identification of contents.
- (3) *Reproducible analysis pipelines:* Our scripts have been documented in smaller version-controlled R Markdown, or R scripts, making it easier to follow the analytical workflow step-by-step.
- (4) *Public repository:* We have uploaded all data, code, and documentation to GitHub. This repository includes clear instructions for replicating the analysis, as well as links to the necessary software and dependencies. The link to the repository will be

made public as soon as the manuscript is accepted for publication. For the purpose of this review, the files can be found on the following link: <https://lu.box.com/s/dcdulv1iw6gyvy0bp6jsya59e1tm5fp6>

Below are my comments on the described data analyses:

1. *PCA showed clustering of cohort and visit, and was used as a basis for confounder selection. Confounders should be selected a-priori and should include all necessary factors regardless of whether the PCA show clustering on these variables or not. Simply stated, PCA is not a tool for confounder selection. It only tells us the a big part of the variability in the data is contributed by cohort and visit. This does not indicate whether a certain variable would influence the relationship between exposure and outcome.*

Response: We agree with Reviewer #1 that confounders should be determined *a priori* based on domain knowledge and relevant literature, rather than solely relying on PCA. We used PCA to detect global patterns and sources of technical variation within the dataset. This preliminary analysis helped us to identify substantial variability attributed to cohort and visit, which informed our understanding of the data structure. To account for these observations, we then decided to adjust the differential expression analysis for principal components (PC1 and PC2).

Based on this and other feedback, we have reanalyzed the data and revised the manuscript accordingly. In summary, confounders were selected *a priori* based on existing literature and biological relevance. Both age and sex are well-known factors influencing HIV-1 progression, whereas cohort and subtype may reflect potential variability due to demographic and virological differences. Next, age, sex, cohort, and HIV-1 subtype were assessed to ensure comprehensive adjustment for potential biases. The assessment showed that the Durban cohort consisted entirely of female participants, whereas the IAVI cohort predominantly included male participants (34/39). Hence, adjusting for cohort also inherently adjusts for sex. Similarly, the Durban cohort was composed exclusively of individuals infected with HIV-1 subtype C, whereas the IAVI cohort mainly was composed of individuals infected with HIV-1 sub-subtype A1 (31/39). Adjusting for cohort therefore also inherently adjusts for HIV-1 subtype. Given these considerations, including sex and subtype as additional covariates in our models would be redundant and result in an over-parameterized model, since their effects are already largely encapsulated by the cohort variable.

To ensure consistency and robustness, the statistical models were therefore designed as follows:

- a. Differences between time-points (V1-V0, V2-V0, V2-V1): Linear mixed model adjusting for age and cohort.
- b. ARS (Acute Retroviral Syndrome): Linear regression adjusting for age (since information on ARS was only available for the IAVI cohort, we could not adjust for cohort in this model).
- c. Viral load: Linear regression adjusted for age and cohort.
- d. Disease progression: Cox regression adjusting for age and cohort.

This has been detailed on p. 6, lines 11-19, and p. 39, lines 5-7 in the revised manuscript, including a comprehensive description of the confounder selection and the statistical rationale for each model. Details of the statistical models and the control for confounders is now also summarised in Supplementary Table S4.

2. *It is also intriguing that the set of confounders differ a lot depending on the analysis. Some are controlled for cohort only (timepoint differences), age only (association with ARS), age + cohort (HIV control), and age+sex+cohort+subtype (Cox regression). It is my opinion that all these (age, cohort, sex and subtype) are potential confounders in all these analyses and should be accounted for. The authors may also consider cohort as a random effect to allow for each cohort to have a different slope and intercept. This may make the analysis better as genetic and environment factors may be very different in the 2 countries.*

Response: As described above (response to comment #1), we have now reanalyzed the data and controlled for confounders in a more standardized way. Regarding Reviewer #1's suggestion to consider cohort as a random effect, we appreciate the potential benefit of allowing each cohort to have different slopes and intercepts due to varying genetic and environmental factors between countries. However, given that our cohort variable only has two levels (Durban and IAVI), we believe that including it as a random effect would be statistically inappropriate. Instead, in the time differences analysis, we have included cohort, as well as the interaction *cohort:visit* as fixed effect variables (p. 8, lines 7-9; p. 41, lines 13-15), while also accounting for differences in baseline logI between individuals through the random intercept. Here is a summary of updated model equations for different analyses:

Differences between time-points (V1-V0, V2-V0, V2-V1):

$\log I \sim \text{age} + \text{visit} + \text{cohort} + \text{visit:cohort} + \text{PC1} + \text{PC2} + (1|\text{patientid})$

ARS * only one cohort accessed

$\log I \sim \text{ARS} + \text{visit} + \text{ARS:visit} + \text{age} + (1|\text{subjid})$

HIV control

$\log I \sim \text{VL} + \text{visit} + \text{VL:visit} + \text{age} + \text{cohort} + (1|\text{subjid})$

After model adjustments with uniform cofounders across all analyses the results as presented in the below table were observed (the results from the previous unadjusted analysis have been added below for comparison). In summary, the top proteins identified in the previous analysis remain differentially expressed at a similar magnitude and statistical support, meaning that the overall conclusions of the study remain.

Differences between time-points (V1-V0, V2-V0, V2-V1)

Protein	Old analysis (unadjusted)	New analysis (adjusted): age IAVI vs. Durban	Interpretation of changes
V10			
VWF	P: 7.1e-9 Q: 3.0e-7	P: 1.1e-8 Q: 3.9e-7	The adjusted model confirms the significance but with a slight change

	FC: 1/1.5 Significant: Yes	FC: 1/1.6 Significant: Yes	in fold change. The association remains strong and consistent.
FN1	P: 1.5e-5 Q: 1.6e-4 FC: 0.4/1.3 Significant: Yes	P: 1.9e-5 Q: 2.1e-4 FC: 0.4/1.2 Significant: Yes	The adjusted model confirms the significance but with a slight change in fold change. The association remains strong and consistent.
V20			
TTN	P: 1.9e-9 Q: 9.2e-8 FC: 2.2/2.2 Significant: Yes	P: 3.9e-9 Q: 1.7e-7 FC: 2.3/2.2 Significant: Yes	The adjusted model confirms the significance but with a slight change in fold change. The association remains strong and consistent.
HINT2	P: 1.5e-5 Q: 1.6e-3 FC: 1.4/1.6 Significant: Yes	P: 2.0e-3 Q: 0.01 FC: 1.4/1.6 Significant: Yes	The adjusted model confirms the significance but with a slight change in fold change. The association remains strong and consistent.
VWF	P: 7.2e-9 Q: 3.0e-7 FC: 1.1/1.9 Significant: Yes	P: 7.2e-9 Q: 3.0e-7 FC: 1.1/1.9 Significant: Yes	The adjusted model confirms the significance but with no change in fold change. The association remains strong and consistent.
FN1	P: 1.4e-5 Q: 1.6e-4 FC: 0.4/1.3 Significant: Yes	P: 1.9e-5 Q: 2.1e-4 FC: 0.5/1.4 Significant: Yes	The adjusted model confirms the significance but with a slight change in fold change. The association remains strong and consistent.
PAPOLA	P: 2.7e-5 Q: 2.9e-4 FC: 0.8/2.0 Significant: Yes	P: 2.7e-5 Q: 2.9e-4 FC: 0.7/2.0 Significant: Yes	The adjusted model confirms the significance but with a slight change in fold change. The association remains strong and consistent.
RAB10	P: 4.6e-5 Q: 4.5e-4 FC: 1.2/1.3 Significant: Yes	P: 4.8e-5 Q: 4.6e-4 FC: 1.2/1.3 Significant: Yes	The adjusted model confirms the significance but with no change in fold change. The association remains strong and consistent.
FLNA	P: 3.4e-8 Q: 1.0e-6 FC: 0.9/1.1 Significant: Yes	P: 0.02 Q: 0.05 FC: 0.9/1.1 Significant: No	The fold change is similar between models but not significant. Protein expression could be affected by age.
HLA-A	P: 7.4e-6 Q: 1.6e-4 FC: 0.8/1.0 Significant: Yes	P: 7.3e-6 Q: 1.0e-4 FC: 0.8/1.0 Significant: Yes	The adjusted model confirms the significance but with a slight change in fold change. The association remains strong and consistent.
LTF	P: 4.2e-11 Q: 3.8e-9 FC: -1.2/-0.8 Significant: Yes	P: 3.8e-11 Q: 3.4e-9 FC: -1.2/-0.8 Significant: Yes	The adjusted model confirms the significance but with a slight change in fold change. The association remains strong and consistent.
PRDX2	P: 1.6e-11 Q: 1.8e-9 FC: -1.3/-1.1 Significant: Yes	P: 1.9e-11 Q: 2.1e-9 FC: -1.3/-1.1 Significant: Yes	The adjusted model confirms the significance but with a slight change in fold change. The association remains strong and consistent.
FGL1	P: 3.0e-7 Q: 6.4e-6 FC: -1.1/-1.4 Significant: Yes	P: 3.9e-7 Q: 7.9e-6 FC: -1.2/-1.4 Significant: Yes	The adjusted model confirms the significance but with a slight change in fold change. The association remains strong and consistent.
ATF6	P: 5.2e-5 Q: 4.8e-4 FC: -1.4/-2.4 Significant: Yes	P: 5.5e-5 Q: 5.1e-4 FC: -1.4/-2.4 Significant: Yes	The adjusted model confirms the significance but with a slight change in fold change. The association remains strong and consistent.
V21			

PI16	P: 1.8e-3 Q: 9.4e-3 FC: 1.2/1.6 Significant: Yes	P: 1.4e-3 Q: 7.74e-3 FC: 1.2/1.6 Significant: Yes	The adjusted model confirms the significance with no difference in fold change. The association remains strong and consistent.
HNRNPA2B1	P: 2.8e-5 Q: 2.9e-4 FC: -1.1/-1.2 Significant: Yes	P: 2.1e-5 Q: 2.3e-4 FC: -1.1/-1.2 Significant: Yes	The adjusted model confirms the significance with no difference in fold change. The association remains strong and consistent.
PRDX2	P: 1.6e-11 Q: 1.8e-9 FC: -1.7/-0.8 Significant: Yes	P: 1.8e-11 Q: 2.1e-9 FC: -1.7/-0.8 Significant: Yes	The adjusted model confirms the significance with no difference in fold change. The association remains strong and consistent.
MANBA	P: 3.2e-3 Q: 1.5e-2 FC: -1.4/-1.5 Significant: Yes	P: 3.3e-3 Q: 1.5e-2 FC: -1.4/-1.5 Significant: Yes	The adjusted model confirms the significance with no difference in fold change. The association remains strong and consistent.
HPN	P: 3.5e-4 Q: 2.5e-3 FC: -1.6/-1.3 Significant: Yes	P: 3.5e-4 Q: 2.5e-3 FC: -1.6/-1.3 Significant: Yes	The adjusted model confirms the significance with no difference in fold change. The association remains strong and consistent.
GRN	P: 3.3e-5 Q: 3.3e-4 FC: -2.6/-1.9 Significant: Yes	P: 3.5e-5 Q: 3.3e-5 FC: -2.7/-1.9 Significant: Yes	The adjusted model confirms the significance with no difference in fold change. The association remains strong and consistent.

ARS

Protein	Old analysis (unadjusted)	New analysis (Adjusted): age + ARS:visit	Interpretation of changes
V10			
ICOSLG	FC: -0.6 P: 0.0017 Significant: Yes	FC: -0.5 P: 0.0029 Significant: Yes	The adjusted model confirms the significance with no difference in fold change. The association remains strong and consistent.
HRG	FC: -0.5 P: 0.0041 Significant: Yes	FC: -0.5 P: 0.0004 Significant: Yes	The adjusted model confirms the significance with no difference in fold change. The association remains strong and consistent.
SCGB1A1	FC: -3.7 P: 0.0007 Significant: Yes	FC: -3.7 P: 0.0038 Significant: Yes	The adjusted model confirms the significance with no difference in fold change. The association remains strong and consistent.
ZYX	FC: -3.9 P: 0.0004 Significant: Yes	FC: -3.1 P: 0.0009 Significant: Yes	The adjusted model confirms the significance with no difference in fold change. The association remains strong and consistent.
ECMI	FC: -0.3 P: 0.04 Significant: No	FC: -0.4 P: 0.002 Significant: Yes	The adjusted analysis reveals that this protein is significantly differentially expressed, which was not captured in the initial analysis.
LILRA3	FC: 1.0 P: 0.02 Significant: No	FC: 1.0 P: 0.001 Significant: Yes	The adjusted analysis reveals that this protein is significantly differentially expressed, which was not captured in the initial analysis.
V20			

ICOSLG	FC: -0.6 P: 0.0005 Significant: Yes	- FC: 0.6 P: 0.0029 Significant: Yes	The adjusted model confirms the significance with no difference in fold change. The association remains strong and consistent.
HRG	FC: -0.5 P: 0.0039 Significant: Yes	FC: -0.5 P: 0.0004 Significant: Yes	The adjusted model confirms the significance with no difference in fold change. The association remains strong and consistent.
SCGB1A1	FC: -3.4 P: 0.0015 Significant: Yes	FC: -3.0 P: 0.0038 Significant: Yes	The adjusted model confirms the significance but with a slight change in fold change. The association remains strong and consistent.
PSMA1	FC: 2.0 0.0005 Significant: Yes	FC: 1.8 0.0011 Significant: Yes	The adjusted model confirms the significance but with a slight change in fold change. The association remains strong and consistent.
GDI1	FC: 3.3 P: 0.0029 Significant: Yes	FC: 3.0 P: 0.0002 Significant: Yes	The adjusted model confirms the significance but with a slight change in fold change. The association remains strong and consistent.
GDI2	FC: 1.6 P: 0.0029 Significant: Yes	FC: 1.3 P: 1.8e-5 Significant: Yes	The adjusted model confirms the significance with a slight difference in fold change. The association remains strong and consistent.
GSTO1	FC: 1.2 P: 0.0010 Significant: Yes	FC: 1.1 P: 0.003 Significant: Yes	The adjusted model confirms the significance with no difference in fold change. The association remains strong and consistent.
ECM1	FC: -0.3 P: 0.105 Significant: Yes	FC: -0.3 P: 0.002 Significant: Yes	The adjusted analysis reveals that this protein is significantly differentially expressed, which was not captured in the initial analysis.

3. *PLS-DA was used for the multivariate analysis of ARS with an average accuracy of 78% assessed through cross-validation. PLS-DA is very sensitive to overfitting, especially in small sample sizes, hence more model parameters are needed to demonstrate that the model is robust. The authors are encouraged to also show the AUROC, misclassification error (confusion matrix shown), and most importantly an external validation (which given the small sample size might not be possible).*

Response: We thank Reviewer #1 for these insightful comments and recommendations for additional metrics and validations to ensure the robustness of the model. Below, we detail the steps and additional analyses that we have taken to address Reviewer #1's suggestions:

(1) *Overfitting and model robustness:* We agree that PLS-DA is sensitive to overfitting, especially with small sample sizes. To mitigate this risk, we have applied a rigorous 10 k-fold cross-validation technique (with k=5) to assess the model's performance, and to ensure that the reported accuracy of 78% is reliable and not a result of overfitting. In summary, the cross-validation splits the data into parts (or "folds"). For example, with k=5, the data is divided into five chunks. The model is trained on four of these parts and tested on the remaining one. This process is repeated five times, so that each part of the data is tested

once. The results are averaged to give an overall performance measure. This method helps preventing the model from being biased or "overfitted" to the training data. While $k=10$ is common and generally more rigorous, $k=5$ still provides a reliable estimate of how well the model will perform on new, unseen data. It is also less computationally intensive than higher folds, making it a good balance between accuracy and efficiency. Finally, and for clarification, the cross-validation was only used for model validation, not for tuning.

(2) *Additional model parameters and metrics:* To comply with Reviewer #1's suggestion and to further assess the robustness of the PLS-DA model, we have now included the following additional metrics to the manuscript:

- a. Area Under the Receiver Operating Characteristic Curve (AUROC): This metric provides a comprehensive measure of the model's discriminative ability and showed an average AUROC of 0.82 (computed over 50 test sets).
- b. Misclassification error and confusion matrix: We have computed the misclassification error and provided the metrics in a confusion matrix to illustrate the model's performance in correctly classifying ARS-positive and ARS-negative samples (p. 10, lines 10-15). In addition, all performance measures of the models have been summarized in Table S5.
- c. Chosen model: v10v20
 - i. ER:0.20
 - ii. Accuracy:0.80
 - iii. AUC:0.82
 - iv. TN:0.26
 - v. TP:0.54
 - vi. FP:0.15
 - vii. FN:0.05
 - viii. Misclassification error = proportion of samples which were incorrectly classified = $1 - \text{accuracy} = 0.2$

(3) *External validation:* We recognize the importance of external validation in establishing the generalizability of our findings. However, given the rarity of linked longitudinal samples collected both prior to infection and during untreated hyperacute HIV-1 infection, it is simply not feasible to conduct a separate and independent external validation of the results. This highlights the importance and uniqueness of our results data providing insights into the human host response during the establishment of an HIV-1 infection at an unprecedented level. Still, we acknowledge the importance of addressing this limitation and have therefore implemented a stratified cross-validation approach to ensure that the training and testing sets are representative of the overall data distribution. This way an external validation scenario can be simulated and provide additional confidence in the model's robustness.

4. *The authors also mentioned that a linear regression model was performed to CONFIRM the association between the identified proteins in the PLS-DA and ARS. I would like to remind the authors that univariable analysis and multivariable analysis serve different purposes and do not confirm each other. PLS-DA should be viewed entirely as a collective contribution of proteins groups. Therefore, proteins that score high in PLS-DA may do so because they are together with other proteins despite not having a substantial effect individually.*

Response: We fully agree with Reviewer #1 that univariable analysis and multivariable analysis serve different purposes and do not confirm each other; and that PLS-DA should be considered as evaluating the collective contribution of protein groups. In accordance, we described the PLS-DA results as identifying groups of proteins that collectively differentiate between ARS-positive and ARS-negative cases. Our intention was not to use linear regression to confirm the results of the PLS-DA, but rather to complement the multivariate analysis by providing additional insight into the relationships between individual proteins and ARS. Having re-read the previous version of the manuscript, we acknowledge that the wording was not entirely clear. We thank Reviewer #1 for bringing this to our attention and have now revised the manuscript to more accurately reflect the purpose of this analysis (p. 41, lines 11-18; p. 10-11, lines 23-25, 1).

5. *After doing PLS-DA and linear regression, the authors follow with a hierarchical clustering to identify the top proteins that distinguished participants with or without ARS. This is a lot of steps to get to the point. It is unclear to me why these steps were taken when the PLS-DA and linear regression can already provide top hits of proteins.*

Response: We thank Reviewer #1 for this comment on the rationale behind our methodological approach. We understand that the multiple steps in our analysis may appear redundant. However, each step serves a distinct purpose in comprehensively identifying and visualizing the proteins associated with ARS. We can consider moving the hierarchical clustering to supplementary documentation.

PLS-DA: The PLS-DA was employed to identify groups of proteins that collectively differentiate participants with and without ARS. This multivariate analysis considers the combined effect of protein expression profiles.

Linear regression: Subsequently, linear regression was used to assess the individual association of each protein with ARS. This step allows us to understand the specific contributions of each protein identified in the PLS-DA.

Hierarchical clustering and heatmap: We agree with Reviewer #1 that the PLS-DA and the linear regression would be sufficient to provide the top hits of proteins. Our reasoning was that the heatmap, generated through hierarchical clustering, might help the reader as a visual tool to illustrate the expression patterns of the top proteins across participants. The method simply cluster both patients and proteins based on their expression levels, showing groups of proteins that exhibit coordinated changes in relation to ARS status. In addition, the heatmap also provides an intuitive and comprehensive visualization of the data, highlighting patterns that may not be

immediately apparent from PLS-DA and linear regression. In short, by including hierarchical clustering and the heatmap, we aimed to provide a clearer and more holistic understanding of the protein expression differences associated with ARS. However, we will leave it up to the editor to decide whether this should be included in Figure 2 or not.

6. *Regarding the Cox regression, the interpretability of the hazard ratios is questionable. Since the protein levels are differences, what do these HRs really mean? A 1 point increase in the difference between V1 and V0 for PRKCB is associated with a 30% increase in risk for getting CD4+ below 500?? This seems difficult to believe. If the data is normalized and/scaled, then the interpretation is ever more tricky. This question relates to one of the main pitfalls when using differences (or change scores) as predictors (or outcomes) in linear regression – interpreting the result beyond saying that the result is “significant”.*

Response: We thank Reviewer #1 for these thoughtful comments. Indeed, the actual protein levels vary between individuals and makes it more informative to compare the changes in protein levels between visits rather than focusing at the individual quantified levels. We used log₂-transformed protein values, meaning that a 1-unit increase between V0 and V1 indicates a doubling of the protein level. Hence, a HR of 1.3 indicates that a doubling of the protein level (a 1-unit increase in the log₂ scale) from V0 to V1 is associated with a 30% increase in risk that CD4+ T-cell counts will drop below 500 within one year from estimated date of infection. We would argue that this interpretation is statistically sound since the log₂ transformation standardizes the data and allows for a meaningful comparison of the relative changes in protein levels. The difference in protein levels (V1-V0) was used to account for individual baseline variability. This approach helps in understanding the impact of the change in protein levels over time on disease progression, rather than absolute levels, which can be influenced by a variety of external factors. Importantly, this was only possible due to the unique longitudinal data available from within the study participants including the pre-infection levels, enabling a more personalized and dynamic view of how these changes unfold in real time, thus minimizing external biases.

However, we appreciate that this interpretation may seem counterintuitive at first glance, but it reflects the proportional nature of the Cox regression model, which assesses the relative risk associated with changes in predictor variables. We have ensured that our data normalization and scaling procedures are appropriately accounted for in the model, making these HRs both meaningful and interpretable within the context of our study. To help the reader, we have now clarified how to interpret the determined HRs in the manuscript (p. 13, lines 18-21; p. 42, lines 15-23). Moreover, we have also added a discussion on the biological plausibility of the associations and the potential clinical implications of our findings (p. 13-14, lines 21-25, 1-4; p. 17-18, lines 24-25, 1-10).

7. *Another issue is regression to the mean. Taking the difference between two variables assumes that all other conditions are the same during those two time points, which is hardly true. There could be external factors (change in diet, infections, for example) that influence the variability in V0 that are no longer there during V1 (vice versa). Hence, V1 – V0 may be heavily confounded. Moreover, when using difference scores,*

regression to the mean can occur if extreme values at one time point tend to be closer to the mean at the subsequent time point. This can lead to an apparent change in the outcome variable that is not related to disease progression.

To mitigate these, the authors are encouraged to also consider performing the analysis with V1 or V2 as the outcome, and adjusting for V0 instead of regressing on the differences. This way, the HRs are more interpretable and we reduce the other problems that are described above.

Response: We thank Reviewer #1 for this comment and appreciate the potential caveats of using difference scores. In compliance with Reviewer #1's suggestion, we have performed several additional analyses to provide more robust results as follows:

- (1) We have instead used interaction terms in the mixed effects model to allow the model to directly capture how changes over time vary by specific factors (e.g., group or condition), without making such assumptions. These interaction terms between time (visit) and other variables like cohort allow the model to consider both within- and between-subject variability. This approach helps differentiate genuine changes in protein expression or clinical outcomes due to the variable of interest (e.g., ARS) from those that could be explained by random variability or regression to the mean. For example, an interaction term between "visit" and "ARS" in the mixed model would account for the trajectory of protein expression over time, considering both the baseline values and the change patterns. This effectively counters the potential distortion caused by extreme baseline values moving closer to the average in subsequent measurements (regression to the mean). However, using V1 or V2 as the outcome, and adjusting for V0 does not directly respond to the question whether protein value changes between visits are related to ARS. Instead, it responds to whether a difference in V1 protein levels between ARS Yes and No are independent of protein levels at V0.
 - (2) We employed Cox proportional hazards models with time to CD4<500 as outcome variable and V10 or V20 or V21 independent variable as well as age and cohort to provide more interpretable hazard ratios. These HRs reflected the association between protein levels at a subsequent visit and the risk of disease progression, while controlling for baseline levels. When using Cox proportional hazards models, interaction terms cannot be effectively employed to account for changes in longitudinal data. This is because the Cox model is designed to evaluate the relationship between survival time and covariates, which typically do not involve repeated measures over time. In such models, interaction terms can introduce challenges related to multicollinearity and overfitting, particularly in the context of time-varying covariates. Instead, employing difference scores (e.g., V1-V0) provides a more practical way to capture longitudinal changes and minimize the risk of regression to the mean.
8. *Another solution could be to use the latent clusters as the exposure instead of the protein levels themselves. The authors have already determined which proteins move*

in the same way. It might help to say that the hazard of disease progression is higher among those that follow pattern A compared to pattern B, and by an HR of this much.

Response: We thank Reviewer #1 for yet another thoughtful suggestion. Latent clusters as the exposure variable may indeed provide a more robust understanding of the relationship between protein expression patterns and disease progression. To comply with this suggestion, we used the latent clusters as predictor variables in a Cox regression model, as opposed to using individual protein levels. Participants were categorized based on in which of the six clusters they clustered (the clustered were based on longitudinal expression profiles), and the hazard ratios (HRs) for disease progression were calculated for each cluster. Hazard ratios (HRs) for disease progression were then calculated for each cluster. This approach allowed us to assess the HRs for disease progression based on different protein expression patterns (e.g., pattern A vs. pattern B). Mixed-effects Cox proportional hazards models were used to account for the correlation between proteins within the same patient.

Formula: `cox_model <- coxme(Surv(time, status) ~ cluster + age + sex + cohort + (1 | patient_id), data = your_data)`

Since the p-values were low and z-values, close to zero for all the cluster comparisons, this suggested that the cluster variable did not have any significant effect on comparing the risk of disease progression across different protein clusters. Moreover, we also assessed proteins expression collectively using PLS-DA models to capture the synergistic effects and interactions between multiple proteins that may not be apparent when examined individually. However, no group of proteins was collectively associated with viral control or disease progression.

9. *Final remarks, the labels in Figure 3 c and e do not make sense. What do V0V1V2, V0V10V20 and others mean?*

Response: We thank Reviewer #1 for pointing out that this was not entirely clear as presented. This figure relates to the assessment of the robustness of the PLS-DA modelling (see also response to comment #3 above). However, this was not entirely clear as written, and we have now revised the figure and the figure legend for clarity.

10. *In summary, I am convinced that this is an important topic. However, I am keen to see an improved data analysis before I can be fully convinced of the results.*

Response: We thank Reviewer #1 for this encouraging comment and appreciate the constructive feedback very much. A robust and thorough analytical approach is essential for the validity of our findings, and we believe that the comments and suggestions have improved the quality and robustness of our study. Please find below a summary list of the revised and implemented analyses:

- (1) *Confounder selection and model consistency:* We have revisited our approach of confounder selection to ensure consistency across all analyses, and all models now include age and cohort as covariates to address potential confounding factors comprehensively. As described, sex and subtype are encapsulated within the cohort

variable, and was therefore not be added independently to avoid redundancy and an over-parameterized model.

- (2) *PLS-DA model validation*: To address concerns about the robustness of our PLS-DA model, we have provided additional model validation metrics – including AUROC and the misclassification error curve and confusion matrix.
- (3) *Linear regression analysis*: We acknowledge the limitations of using change scores in linear regression due to potential confounding factors and regression to the mean. We have therefore performed an additional analysis using V1 or V2 as the outcome, while adjusting for V0 to mitigate the effect of regression to the mean difference. In the Cox model, we used either V1 or V2. This method will provide more interpretable hazard ratios (HRs) and mitigate some of the issues associated with difference scores.
- (4) *Enhanced data analysis section*: We have revised the data analysis section of our manuscript to include detailed model descriptions, predictor and dependent variables, covariates, and employed statistical methods. This will ensure that our analysis is transparent and reproducible.

In conclusion, we believe that the clarifications and additional analysis added to the revised manuscript have enhanced the robustness of the results and strengthened the overall conclusions.

Comments from Reviewer #2

Comments to the Author

Jamirah Nazziwa et al., reported a comprehensive longitudinal study of proteomes (DIA-MS) from precious retrospective plasma samples collected before, during, and after hAHI in 57 HIV infected participants from four sub-Saharan African countries. The in silico analysis of the proteome results have been associated with different outcomes including: i) ARS in hyperacute HIV-1 infection, ii) Virus control and iii) risk of disease progression. Then a longitudinal dynamic of key factors has been also evaluated. From this analysis different candidates have emerged, such us: Zyxin (ZYG), Secretoglobin family 1A member 1 (SCGB1A1), and Pro-platelet basic protein (PPBP) for ARS; Rho GTPase activating protein 18 (ARHGAP18), Annexin A1 (ANXA1), and Lipopolysaccharide binding protein (LBP) for viral loads; and Hepsin (HPN), Protein kinase C beta (PRKCB), and Integrin subunit beta 3 (ITGB3) for disease progression.

Response: We thank Reviewer #1 for appreciating and acknowledgement of the comprehensive analysis and uniqueness of the samples included in this large collaborative effort.

- 1. Given the proximity of V0 to seroconversion, I was wondering if you have assessed reservoir levels at peripheral blood. In this regard, do you have such information for later points?*

Response: We thank Reviewer #2 for the comment regarding reservoir levels in peripheral blood. This is an interesting question, but unfortunately, we do not have access to such data. In addition and related to the rarity and uniqueness of the samples, we do not have access to cells to assess this retrospectively.

- 2. Durban vs IAVI cohorts display market differences....age, gender, subtype, depletion vs non protein depletion, instrument DIA/MS...and it is not clearly explained the control for cofounders and co-variates in each analysis performed. In that sense, aside from the many sources of variability (differences in sex, age, subtype, African region, protein depletion vs non-depletion, instrument etc...) different models have been applied PLS-DA, Mixed linear model (Age, cohort), COX model (age, sex, cohort, substudy) being very confusing the rationale and the argument behind the selection of the model and the variable controls in each of them as well as samples used. It is recommended to detail this information in data analysis section in methods.*

Response: We thank Reviewer #2 for this important comment. To comply with this comment, we have made significant revisions to the Methods and Results sections to increase the clarity of how potential confounders have been dealt with throughout the study.

In brief, to account for cohort differences, confounders have now been selected *a priori* based on existing literature and biological relevance. Both age and sex are well-known factors influencing HIV-1 progression, whereas cohort and subtype may reflect potential variability due to demographic and virological differences. Next, age, sex, cohort, and HIV-1 subtype was assessed to ensure comprehensive adjustment for potential biases. The

assessment showed that the Durban cohort consisted entirely of female participants, whereas the IAVI cohort predominantly included male participants (34/39). Hence, adjusting for cohort also inherently adjusts for sex. Similarly, the Durban cohort was composed exclusively of individuals infected with HIV-1 subtype C, whereas the IAVI cohort mainly was composed of individuals infected with HIV-1 sub-subtype A1 (31/39). Adjusting for cohort therefore also inherently adjusts for HIV-1 subtype. Given these considerations, including sex and subtype as additional covariates in our models would be redundant and result in an over-parameterized model, since their effects are already largely encapsulated by the cohort variable. The manuscript has now been revised to clearly outline how confounders and covariates were controlled for in each analysis (p. 6, lines 11-19; p. 39, lines 5-7). We have also included details on the methods used to identify and control for these confounders in our statistical models as summarised in Supplementary Table S4.

Regarding potential technical variability related to system migration and analysis of neat samples vs. samples in which the 14 most abundant proteins had been depleted. Multiple *in-house* evaluations was performed at the core facility for mass spectrometry at Lund University ensuring the robustness and comparability of both instruments used.

This did not affect the number/intensity of proteins quantitated. Details of the specific plasma preparation, LC-MS/MS run conditions, instrumentation and spectral library are presented in Supplementary methods. Moreover, as acknowledged in the limitations section of the revised manuscript, the overall quantification of proteins increased 3-fold by the introduction of depletion columns. However, and importantly, the majority of proteins that were detected in both the neat and depleted approaches were comparable (p. 18-19, lines 22-25, 1-2). Details of the specific plasma preparation, LC-MS/MS run conditions, instrumentation and spectral library are presented in Supplementary methods.

3. *HIV controllers and rapid/slow progressors information: these participants have been HLA typed ? There is information as well for CCR5 delta 32?*

Response: We thank Reviewer #2 for the comment regarding HLA typing and characterization of the CCR5 delta 32 variant. Information on the CCR5 delta 32 variant has not been collected for the study participants. However, HLA typing has been performed for 53/54 of the study participants. The detailed HLA allele information, including HLA-A, HLA-B, and HLA-C loci, has now been included in Supplementary Table 1. Although we consider this is out of the scope of this study, we have added an analysis of associations between HLA types and HIV control and disease progression (p. 6, lines 19-23; p. 42, lines 11-13). However, no associations were found between HLA and HIV control or disease progression.

4. *Many genes are mentioned in the text in abbreviated form (gene ID) without first mentioning their full names*

Response: We thank Reviewer #2 for this remark and have now revised the manuscript to ensure that all gene abbreviations are introduced with their full names upon first mention. For example, where we initially mentioned "CXCL10" without introducing this abbreviation, we have now updated the text to "C-X-C motif chemokine ligand 10 (CXCL10)" throughout the revised manuscript.

5. *There is a lack of study limitations in the discussion section considering all the variability sources and how the gender, age, African region, subtype information, technical performance that needs to be more explained.*

Response: We thank Reviewer #2 for this comment and have now expanded the Discussion section to include a thorough discussion of the study's limitations, including cohort differences and technical variability (p. 18-19, lines 15-25; 1-3).

6. *The results obtained in this study are detailed descriptively, and although there is a literature search for the molecules found to be relevant in this study and they are framed in the context of HIV, the findings are not discussed in depth in the clinical context outlined in the study's objectives (associations with ARS, viral load responses, and HIV-1 disease progression) and the impact they have or could have as blood biomarkers and/or targets for prophylactic and therapeutic HIV-1 interventions*

Response: We thank Reviewer #2 for this valuable feedback of providing a more in-depth discussion of our findings from a clinical context, specifically regarding associations with Acute Retroviral Syndrome (ARS), viral load responses, and HIV-1 disease progression. In the revised manuscript, we now elaborate on the potential impact of the identified top proteins as blood biomarkers and/or targets for prophylactic and therapeutic interventions in the revised Discussions section.

7. *An attempt has been made to evaluate the relationship between identified molecules (correlations, ontologies, co-expression, gene neighborhood, protein homology...)? In fact, it would be interesting to see how those molecules that depict the same longitudinal dynamics during hyperacute HIV-1 infection (stormers and slumpers), even if their outcome relationship is different (ARS, viral load, or disease progression), could be related. This way, the molecules would not be treated independently but rather holistically as intended.*

Response: To address the important point about treating molecules with similar longitudinal dynamics holistically during hyperacute HIV-1 infection, we can leverage the over-representation analysis (ORA) performed on the different groups of proteins, as summarized in Figure 3b. By focusing on how these proteins behave collectively rather than independently, we can highlight shared biological pathways and mechanisms that are likely driving their longitudinal dynamics, even if their clinical outcomes (e.g., ARS, viral load, or disease progression) differ.

For instance, proteins grouped into specific longitudinal clusters like the "stormers" (rapidly increasing proteins) and "slumpers" (rapidly decreasing proteins) were analyzed using Gene Ontology Biological Processes (GO-BP) to assess their collective functions. The enrichment analysis identified common processes such as immune response, cellular motility, apoptosis, and viral entry. These shared biological functions suggest that despite variations in outcomes, proteins with similar dynamics likely act in concert to regulate host responses to HIV-1 infection, affecting both viral control and disease progression.

By integrating ORA results, we have now determined how proteins in the same longitudinal cluster contribute to the same or related pathways, and provided a more comprehensive discussion of their role in HIV pathogenesis. These results have been summarised in the extended Fig. 2. Additionally, for outcome variables such as viral control and disease progression, treating proteins holistically proved ineffective, as demonstrated by the results from the PLS-DA models. The models indicated that this approach failed to distinguish between viral controllers and non-controllers, as well as between fast and slow progressors.